# INTERACTSCIENCE: PROGRAMMATIC AND VISUALLY-GROUNDED EVALUATION OF INTERACTIVE SCIENTIFIC DEMONSTRATION CODE GENERATION

## ABSTRACT

Large Language Models (LLMs) are increasingly capable of generating complete applications from natural language instructions, creating new opportunities in science and education. In these domains, interactive scientific demonstrations are particularly valuable for explaining concepts, supporting new teaching methods, and presenting research findings. Generating such demonstrations requires models to combine accurate scientific knowledge with the ability to implement interactive front-end code that behaves correctly and responds to user actions. This capability goes beyond the scope of existing benchmarks, which typically evaluate either knowledge question answering without grounding in code or static web code generation without scientific interactivity. To evaluate this integrated ability, we design a hybrid framework that combines programmatic functional testing to rigorously verify interaction logic with visually-grounded qualitative testing to assess rendered outputs against reference snapshots. Building on this framework, we present INTERACTSCIENCE, a benchmark consisting of a substantial set of carefully designed questions across five scientific domains, each paired with unit tests, reference snapshots, and checklists. We evaluate 30 leading open- and closed-source LLMs and report results that highlight ongoing weaknesses in integrating domain knowledge with interactive front-end coding. Our work positions INTERACTSCIENCE as the first benchmark to automatically measure this combined capability with realistic interactive operations, providing a foundation for advancing reliable and educationally useful scientific demonstration code generation.

## 1 INTRODUCTION

Recent advancements in Large Language Models (LLMs) are catalyzing a fundamental shift in the paradigm of software creation, moving from a process of writing low-level, imperative code to one of articulating high-level, declarative goals (Comanici et al., 2025; OpenAI, 2025). Users now specify a desired outcome (such as "create a tool to visualize protein folding" or "build an interactive simulation of planetary orbits") and expect the LLM to translate this intent into a complete functional application (Chen et al., 2025). This evolving human-AI collaboration is poised to accelerate scientific research and education, empowering scientists to rapidly prototype data visualizations or educators to create bespoke interactive teaching modules, all articulated through natural language (Chu et al., 2025; Van Noorden & Perkel, 2023; Gottweis et al., 2025; Bai et al., 2025a; Sun et al., 2025c). Success is increasingly measured by how well the generated application produces a **functionally correct, visually intuitive, and interactive experience** that faithfully captures the users' intended goals (Sun et al., 2024; Jiang et al., 2024).

In this new paradigm, we focus on the task of **Scientific Demonstration Code Generation** (Ji et al., 2025). These demonstrations are not just static diagrams, they are interactive tools that bring abstract concepts to life, widely used in research and education for explaining complex scientific principles, supporting new forms of teaching, and presenting research findings. This task requires a model to translate abstract scientific principles into a tangible, interactive, and functionally correct system (Ji et al., 2025; Chen et al., 2025). However, this ambitious task exposes a critical limitation of current LLMs that we observed in practical use. For example, as shown in Figure 1, a state-of-the-art LLM can easily explain Newton's second law or generate code for a blog webpage with standard UI

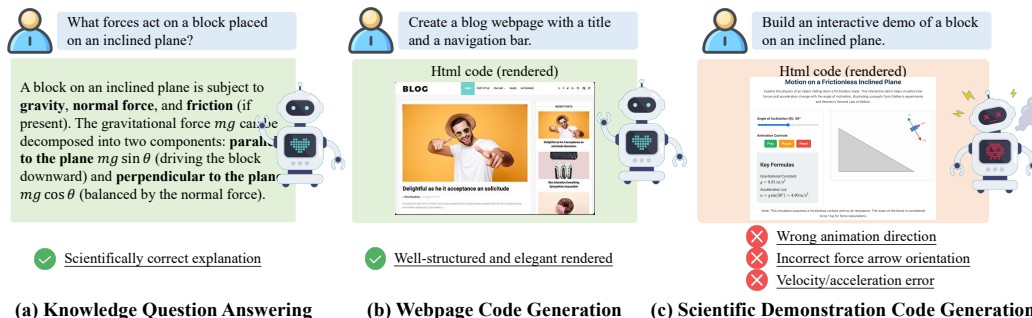

**(a) Knowledge Question Answering**  **(b) Webpage Code Generation**  **(c) Scientific Demonstration Code Generation**

Figure 1: Illustration of three tasks. (a) **Knowledge Question Answering**: given the query about forces act of a block placed on an inclined plane, an LLM can output a correct textual explanation. (b) **Webpage Code Generation**: given the instruction of write a blog webpage, an LLM can generate functional static HTML code. (c) **Scientific Demonstration Code Generation**: generating an interactive demo for the inclined plane scenario, an LLM often fail to produce correct results.

elements. Yet, when asked to combine these skills to generate an interactive web demonstration of a block on an inclined plane, most models fail, producing errors ranging from incorrect physics logic in the JavaScript to non-functional UI components. *This highlights a fundamental gap: models can perform individual tasks but struggle to integrate them effectively* (Feng et al., 2025; Li et al., 2023).

At the same time, existing evaluation methodologies are insufficient for scientific demonstration code generation (Laskar et al., 2024; Chen et al., 2024). Current benchmarks either focus on knowledge question answering (Rein et al., 2024; Hendrycks et al., 2021) or static web code generation (Yun et al., 2024; Gui et al., 2025; Lu et al., 2025), but rarely assess the combination of functional interactivity and scientifically accurate visualization required in interactive demonstrations. Specifically, they lack reliable mechanisms to verify whether user actions correctly trigger the intended scientific behavior, relying on fixed-interval screenshots (Zhang et al., 2025) or element-existence checks (Xiao et al., 2025a) without actual interactions. Moreover, vision-based evaluations that use Vision-Language Models (VLMs) as judges (Gu et al., 2024; Li et al., 2024), typically without reference snapshots, tend to produce subjective judgments that fail to ensure scientific fidelity (Ji et al., 2025). These gaps make it difficult to measure whether a model successfully translates abstract scientific principles into a fully functional, interactive application.

To overcome these limitations, we design a hybrid evaluation framework that combines two complementary components. **Programmatic Functional Testing (PFT)** introduces deterministic unit test verification of interaction logic, ensuring that user actions trigger the intended behavior. **Visually-Grounded Qualitative Testing (VQT)** leverages target snapshots as visual oracles, providing grounded references for VLM-as-judge and enabling reliable assessment of visual correctness. Together, these two components form a robust methodology for evaluating scientific demonstration code. Building on this framework, we construct a new benchmark named INTERACTSCIENCE. It comprises a set of challenging problems across five diverse scientific disciplines: mathematics, physics, chemistry, earth science, and computer science. Each problem is accompanied by a complete evaluation suite, including unit test scripts for programmatic user behavior simulation and verification, reference snapshots for visually-grounded assessment of scientific correctness, and checklists for guidance of VLM-based semantic judgement. To probe the capabilities of current models, we conduct a large-scale evaluation of 30 prominent open- and closed-source models on INTERACTSCIENCE and provide an in-depth analysis of their performance. Our contributions can be summarized as follows:

1. We design a hybrid evaluation framework for the task of scientific demonstration code generation, combining programmatic functional testing with visually-grounded qualitative assessment.

2. We construct and release the INTERACTSCIENCE benchmark, which includes a complete evaluation suite with unit test scripts, reference snapshots, and checklists.

3. We conduct extensive experiments on a wide range of state-of-the-art LLMs and provide a detailed analysis of their performance.

## 2 RELATED WORK

**LLMs for Scientific Visualization.** Recent work has extended LLM evaluation to scientific and educational contexts. Benchmarks like SridBench (Chang et al., 2025) focus on generating scientific figures with semantic and structural accuracy, while EduVisBench (Ji et al., 2025) assesses pedagogically effective visual explanations for STEM education. These approaches emphasize domain knowledge but largely consider static visuals and do not evaluate interactive code or functional correctness. Other efforts treat interfaces as first-class outputs: Chen et al. (2025) shows that LLMs can synthesize task-specific UIs with strong human preference, and Ku et al. (2025) generates theorem explanations using Manim animations with automated metrics. However, such evaluations focus on presentation quality or user perception rather than verifying event-driven correctness in executable, web-based scientific demonstrations. Due to the difficulty of assessing interactive behavior, most prior efforts still rely heavily on manual evaluation. INTERACTSCIENCE *fills this gap by providing an automated evaluation framework with faithful real-interaction simulation, jointly assessing visual quality and scientific correctness.*

**Evaluation of Visual Code Generation.** Much prior work evaluates design-to-code generation using paired datasets or curated benchmarks (Yun et al., 2024; Laurençon et al., 2024; Gui et al., 2025; Si et al., 2025; Sun et al., 2025a; Xiao et al., 2025b; Awal et al., 2025), primarily focusing on static layout fidelity rather than verifying interactive behavior. Some benchmarks, such as Interaction2Code (Xiao et al., 2025a), ArtifactsBench (Zhang et al., 2025), and WebGen-Bench (Lu et al., 2025), consider interactivity using screenshots or basic functional tests, but they often rely on heuristics or subjective VLM/LLM scoring, which can miss subtle event-driven bugs and limit reproducibility. Other code generation benchmarks are summarized in Appendix A. INTERACTSCIENCE *overcomes the limitations of prior benchmarks by providing deterministic functional tests for interactive behavior, rather than relying solely on heuristic or subjective visual assessments.*

## 3 EVALUATION FOR SCIENTIFIC DEMONSTRATION CODE GENERATION

### 3.1 TASK DEFINITION

**Scientific Demonstration.** A **Scientific Demonstration** is an interactive web application with two coupled sections: a **control panel** containing UI elements (e.g., sliders, buttons, inputs) for parameter manipulation, and a **visualization canvas** (e.g., chart, animation, simulation) that dynamically renders the corresponding scientific concept. Its core lies in the interaction logic that binds controls to the canvas, ensuring that visual updates correctly reflect the underlying principles.

**Scientific Demonstration Code Generation.** In this work, the **Scientific Demonstration Code Generation** task is formalized as the creation of such demonstrations: given a detailed **Implementation Plan** $P$, which specifies the page structure, HTML components, initial states and parameters, interaction logic, and visualization techniques, the goal is to generate a self-contained **Front-end Code** artifact $C$. This output is a single HTML file embedding CSS and JavaScript, which must render in a browser as a fully functional demonstration without external dependencies. By framing the task this way, we directly link the design specification $P$ to the resulting functional demonstration, highlighting the dual evaluation requirements of code fidelity and scientific correctness.

### 3.2 PROGRAMMATIC FUNCTIONAL TESTING

Programmatic Functional Testing (PFT) provides deterministic verification of the component definitions in $P$, objectively measuring whether the generated code $C$ behaves as specified.

**Formalism.** A PFT test case is an ordered sequence of action–assertion pairs

$$t_{\text{pft}} = \{(a_i, e_i)\}_{i=1}^N,$$

where each action $a_i$ is a simulated user interaction (e.g., a button click) and each assertion $e_i$ is a predicate on the expected DOM state. The complete test set $T_{\text{pft}}(P)$ for a problem consists of as many unit tests $t_{\text{pft}}$ as interactive components in $P$.

**Execution and Scoring.**   An evaluation function

$$f_{\text{pft}}(C, t_{\text{pft}}) \to \{0, 1\}$$

executes the actions in $t_{\text{pft}}$ on $C$. It returns 1 if all assertions $e_i$ hold, and 0 otherwise, providing an unambiguous measure of functional reliability.

### 3.3 VISUALLY-GROUNDED QUALITATIVE TESTING (VQT)

Visually-Grounded Qualitative Testing (VQT) evaluates the correctness of the visualization and the visual quality of the generated demonstration, anchoring the assessment in explicit visual evidence.

**Formalism.**   A VQT test case is a visual oracle

$$t_{\text{vqt}} = (A, i_{\text{ref}}, L),$$

where $A = (a_1, \ldots, a_k)$ is a sequence of user actions designed to reproduce the state depicted in the reference snapshot, $i_{\text{ref}}$ is that corresponding reference snapshot, and $L = \{l_1, \ldots, l_m\}$ is a checklist of inspection points. The complete test set $T_{\text{vqt}}(P)$ for a problem consists of as many unit tests $t_{\text{vqt}}$ as reference snapshots provided for $P$.

**Execution and Scoring.**   An evaluation function

$$f_{\text{vqt}}(C, t_{\text{vqt}}) \to (\text{CLIP Score, VLM-Judge Score})$$

executes the action sequence $A$ on the generated code $C$. The final action in this sequence is to capture the screenshot, producing the generated snapshot $i_{\text{gen}}$. The function then returns two complementary scores: **Perceptual Similarity**, computed as $\text{CLIP}(i_{\text{gen}}, i_{\text{ref}})$ to measure low-level visual similarity, and **Semantic Correctness**, computed as $\text{VLM}(i_{\text{gen}}, i_{\text{ref}}, L)$ to judge higher-level features guided by the checklist. These scores provide distinct perspectives on visual quality.

## 4 INTERACTSCIENCE BENCHMARK

### 4.1 BENCHMARK COMPOSITION

Each problem instance in the INTERACTSCIENCE benchmark is an evaluation suite containing three components: an implementation plan, a set of unit test scripts, and a set of evaluation checklists for the VLM-as-Judge.

**Implementation Plan.**   Each benchmark problem is defined by a structured implementation plan with five parts: **(1) Page Content Structure.** Defines the logical UI sections (e.g., title, control panel, graph area, formula display) and their roles. **(2) HTML Components.** Lists required HTML elements for each section (e.g., `<div>`, `<canvas>`) and notes libraries if needed. **(3) Component IDs and State.** Assigns each interactive element a unique ID and specifies default values, ranges, steps, and labels or tooltips. **(4) Interaction Logic.** Details how controls affect the application, including DOM updates, formula recalculations, visual re-rendering, and dependencies. **(5) Visualization Techniques.** Specifies rendering methods (e.g., D3.js, Plotly.js, MathJax) and indicates which visuals require real-time updates or animations.

**Test Cases and Unit Test Scripts.**   Derived from the implementation plan, each problem includes a suite of executable test cases to enable our hybrid evaluation framework. As described in Section 3, these are divided into two types. For PFT, we provide scripts composed of an alternating sequence of actions (e.g., simulating a button click) and assertions (e.g., verifying that a text value has updated correctly). For VQT, we provide separate, action-only scripts designed to reproduce the state shown in a target snapshot, culminating in a screenshot command. All test cases are provided as ready-to-run scripts in the Playwright [1] framework.

---

[1] https://playwright.dev

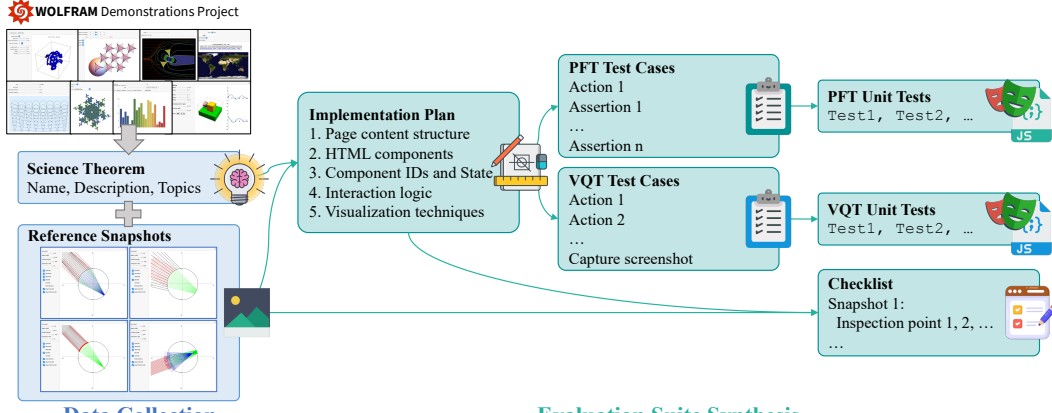

Figure 2: Pipeline of data collection and evaluation suite synthesis. The data collection step retrieves metadata of scientific demonstrations and corresponding snapshots from the Wolfram Demonstrations Project as seed data. The evaluation suite synthesis step generates implementation plans, test cases, unit tests, and checklist sequentially from the seed data.

**Checklists for VLM-as-Judge.** As described in 3.3, each reference snapshot is paired with an evaluation checklist. This checklist is generated from the implementation plan and the underlying scientific principles of the task. It provides a rubric-based guide for the VLM judge, directing it to verify specific points of correctness. These points may include the accuracy of numerical values displayed, the proper alignment of graphical elements, the correctness of labels and legends, and the overall fidelity to the scientific concept being demonstrated. This anchors the VLM's assessment in the ground-truth specifications, moving beyond a purely open-ended visual interpretation.

## 4.2 BENCHMARK CURATION

**Data Collection.** Our benchmark data is sourced from the *Wolfram Demonstrations Project*[2] , which hosts over 13,000 interactive Wolfram Language notebooks. Unlike prior work relying on static materials like textbook (Ji et al., 2025), these executable demonstrations provide natural reference snapshots, ensuring reliable visual ground truth. From this corpus, we manually select 150 demonstrations across five disciplines, with the scale determined by the construction effort and the consideration of maintaining an acceptable evaluation cost (see Appendix C.2). To ensure diversity, we stratify by difficulty, defined by the number of interactive components: 1–3 (**easy**), 4–6 (**medium**), and 7–10 (**hard**), reflecting increasing interaction and visual complexity. For each demonstration, we collect metadata including title, description, topics, and associated snapshots.

**Evaluation Suite Synthesis.** As illustrated in Figure 2, starting from each demonstration's metadata and associated reference snapshots as seeds, we employ the state-of-the-art Gemini-2.5-Pro model to synthesize the corresponding implementation plans, test cases, unit test scripts, and evaluation checklists. The specific prompts used for synthesis are provided in the Appendix E. After each synthesis step, we apply manual inspection and rule-based validation to detect and correct obvious errors, ensuring that each test script is executable. In addition, we conduct a development experiment to verify the quality of the synthesized evaluation suite, as detailed in Appendix B.

## 4.3 BENCHMARK STATISTICS

The INTERACTSCIENCE benchmark consists of 150 problems distributed equally across five scientific disciplines: **Mathematics**, **Physics**, **Chemistry**, **Earth Science**, and **Computer Science**. Each domain contains 30 problems, which are further divided into 10 easy, 10 medium, and 10 hard tasks, resulting in 50 problems for each difficulty level. As detailed in Table 1, the benchmark's statistics show a clear correlation between assigned difficulty and complexity. From easy to hard, there is

---

[2]https://demonstrations.wolfram.com/

Table 1: Statistics of INTERACTSCIENCE benchmark, where **Plan Len.** is the average number of plan tokens, **#Cases** the average test cases, **#Act.** the average actions, **#Asrt.** the average assertions, and **#Check.** the average number of points in checklists, all computed per problem.

| Diff. | #Prob. | Plan Len. | PFT | | | VQT | | |
|---|---|---|---|---|---|---|---|---|
| | | | #Cases | #Act. | #Asrt. | #Cases | #Act. | #Check. |
| Easy | 50 | 2055.84 | 2.54 | 5.68 | 11.36 | 3.98 | 6.44 | 21.56 |
| Medium | 50 | 2320.98 | 3.88 | 9.98 | 19.92 | 3.96 | 9.48 | 21.70 |
| Hard | 50 | 2586.34 | 5.68 | 15.40 | 30.64 | 4.00 | 13.74 | 23.02 |
| **Overall** | **150** | **2321.05** | **4.03** | **10.35** | **20.64** | **3.98** | **9.89** | **22.09** |

a consistent rise in the implementation plan length and the rigor of the evaluation suite, demanding more PFT test cases, user actions, and logical assertions. Compared with prior work (Zhang et al., 2025; Ji et al., 2025), our input plans are substantially longer because they are structured design specifications rather than brief textual hints. The number of VQT test cases, however, remains stable across difficulties because each case corresponds to a reference snapshot, and nearly every problem is equipped with four snapshots to test visual states.

## 5 EXPERIMENTS

### 5.1 EXPERIMENTAL SETUP

**Models.** We evaluate a broad range of state-of-the-art LLMs, including 10 closed-source and 20 open-source models. On the closed-source side, we include the **GPT** (Achiam et al., 2023; Hurst et al., 2024) series, the **Gemini-2.5** (Comanici et al., 2025) series, and the **Claude** series, which represent the most widely adopted commercial models. On the open-source side, we test **DeepSeek-V3** (Liu et al., 2024) and **DeepSeek-R1** (Guo et al., 2025), **Kimi-K2** (Team et al., 2025), **GLM-4.5** (Zeng et al., 2025), **Intern-S1** (Bai et al., 2025a), the **GPT-OSS** (Agarwal et al., 2025) series, the **Qwen3** (Yang et al., 2025a) series, the **Qwen2.5-Coder** (Hui et al., 2024) series, the **Qwen2.5-VL** (Bai et al., 2025b) series, and the **Llama-3.1** series.

**Metrics.** As described in Section 3, we evaluate models along two dimensions. For PFT, we report three pass-rate metrics: the **Overall Pass Rate** (micro average), which is the percentage of unit tests passed across the entire benchmark; the **Average Pass Rate** (macro average), which averages the pass rates over problems; and the **Perfect Pass Rate**, the proportion of problems for which all unit tests pass. For VQT, we also consider three aspects. The **Action Success Rate** measures the percentage of cases where the expected visual state appears after the specified action sequence. Perceptual similarity corresponds to the **CLIP** score between the generated snapshot and the reference snapshot. Semantic correctness is assessed by **VLM-Judge** score, specifically Gemini-2.5-Pro, which checks whether the visual result aligns with the task specification.

Implementation and evaluation details are presented in Appendix C.

### 5.2 RESULTS

**Main Results on Scientific Demonstration Code Generation.** Table 2 summarizes the performance of all evaluated models on INTERACTSCIENCE. Despite differences across models, the absolute performance levels demonstrate that the benchmark is highly challenging. PFT scores remain modest, with the Perfect Pass Rate (PPR) rarely exceeding 16%, underscoring the difficulty of generating code that flawlessly follows the implementation plan. On the visual side, Action Success Rate (ASR) scores are consistently high, often above 85%. However, this metric primarily reflects surface-level interactivity; ASR is high because most models can reliably generate the specified UI components, allowing actions like "clicking a button" or "moving a slider" to execute without error, regardless of whether the resulting visualization is scientifically correct. This superficial competence does not transfer to deeper semantic fidelity. CLIP scores remain moderate and VLM-Judge scores are typically below 60, indicating that while models can create plausible-looking interfaces,

Table 2: Main results of 10 closed-source and 20 open-source models on the INTERACTSCIENCE benchmark. CLIP and VLM-judge scores are normalized to a 0–100 scale.

| Model | PFT | | | VQT | | |
|---|---|---|---|---|---|---|
| | Overall (%) | Average (%) | Perfect (%) | Action (%) | CLIP | VLM-judge |
| *Closed-Source Large Language Models* | | | | | | |
| GPT-5 | 39.47 | **37.61** | **16.08** | 89.66 | 71.95 | **57.02** |
| GPT-4.1 | 37.07 | 34.08 | 11.19 | 89.15 | 71.21 | 52.84 |
| GPT-4o | 28.27 | 27.09 | 5.59 | 85.93 | 67.11 | 42.45 |
| o3 | 34.93 | 32.09 | 13.99 | 89.83 | 72.24 | 52.82 |
| o4-mini | 37.33 | 34.90 | 13.29 | 88.64 | 71.79 | 51.90 |
| Gemini-2.5-Pro | 35.33 | 34.62 | 11.19 | 86.78 | 70.65 | 54.69 |
| Gemini-2.5-Flash | 31.60 | 31.07 | 10.49 | 86.95 | 69.59 | 49.34 |
| Claude-Sonnet-4-20250514 | **41.47** | 37.40 | 13.29 | 89.66 | 73.50 | 55.42 |
| Claude-Opus-4-20250514 | 40.27 | 36.34 | 11.19 | 89.32 | **73.22** | 54.93 |
| Claude-3.5-Sonnet | 33.33 | 31.45 | 9.79 | **90.17** | 72.32 | 49.43 |
| *Open-Source Large Language Models* | | | | | | |
| DeepSeek-R1-0528 | **33.87** | **32.02** | 8.39 | 88.31 | 69.54 | 49.46 |
| DeepSeek-V3-0324 | 31.73 | 30.57 | 10.49 | 85.93 | 68.68 | 49.46 |
| Kimi-K2 | 31.60 | 31.22 | 9.79 | 87.29 | 70.11 | **50.04** |
| GLM-4.5 | 29.33 | 26.65 | 8.39 | 70.51 | 55.90 | 38.57 |
| Intern-S1 | 31.87 | 28.93 | 7.69 | 87.46 | 68.74 | 45.27 |
| gpt-oss-120b | 28.00 | 27.78 | 9.79 | 90.85 | **72.13** | 49.57 |
| gpt-oss-20b | 15.20 | 12.97 | 3.50 | 80.51 | 54.68 | 21.40 |
| Qwen3-235B-A22B-Instruct-2507 | 33.33 | 31.46 | **13.29** | 78.14 | 70.02 | 45.14 |
| Qwen3-32B | 27.20 | 24.09 | 5.59 | 87.46 | 66.46 | 39.69 |
| Qwen3-14B | 24.13 | 23.58 | 7.69 | 85.08 | 66.46 | 36.53 |
| Qwen3-8B | 20.00 | 18.85 | 4.20 | 81.53 | 64.13 | 34.67 |
| Qwen3-4B | 14.67 | 13.10 | 2.80 | 82.03 | 60.90 | 28.33 |
| Qwen3-1.7B | 6.53 | 6.22 | 1.40 | 75.76 | 59.65 | 20.33 |
| Qwen2.5-Coder-32B-Instruct | 27.20 | 25.10 | 7.69 | 84.58 | 51.67 | 38.51 |
| Qwen2.5-Coder-14B-Instruct | 22.53 | 20.61 | 4.90 | 85.42 | 64.47 | 35.72 |
| Qwen2.5-Coder-7B-Instruct | 12.40 | 10.51 | 0.70 | 82.37 | 65.17 | 26.97 |
| Qwen2.5-VL-72B-Instruct | 23.73 | 22.82 | 6.99 | 87.12 | 64.33 | 37.30 |
| Qwen2.5-VL-7B-Instruct | 7.47 | 6.72 | 0.70 | 70.00 | 49.49 | 20.41 |
| Llama-3.1-70B-Instruct | 18.67 | 18.04 | 4.90 | 88.64 | 59.56 | 33.36 |
| Llama-3.1-8B-Instruct | 11.33 | 10.16 | 3.50 | 80.00 | 65.42 | 22.75 |

they often fail to connect these designs with correct physical logic or scientific knowledge. *The gap between ASR and the semantic metrics thus reflects a key limitation: models handle generic frontend generation well but struggle to integrate domain-specific reasoning into functional visualizations.*

Closed-source models generally outperform open-source models, especially in functional correctness, with Claude-Sonnet-4 and GPT-5 achieving the highest PFT scores, indicating stronger adherence to complex implementation plans. Among open-source models, larger ones such as DeepSeek-R1-0528 and Qwen3-235B-A22B-Instruct-2507 reach PFT scores comparable to mid-tier proprietary models, while smaller models ($\leq$14B parameters) struggle on both functional and visual tasks, highlighting the importance of model scale for this synthesis-heavy task. *In short, stronger proprietary models lead in both functional reliability and visual correctness, while scaling is the main driver of performance among open-sourced models.*

Based on these findings, we recommend specific metrics for each evaluation dimension. For PFT, the **Overall Pass Rate (OPR)** is the preferred metric, as it reflects a model's ability to follow functional instructions by measuring adherence to all logical assertions in the plan. For VQT, the **VLM-Judge** score is the most informative indicator, as it directly captures the semantic and scientific correctness of the visualization. The **CLIP** score remains a useful lightweight proxy for perceptual similarity.

**Comparison Across Difficulty Levels and Disciplines.** Figure 3 shows performance across difficulty levels. For PFT, scores remain relatively stable and are sometimes highest on hard problems, reflecting that models can follow instructions for generating interactive controls consistently regardless of the number of components. In contrast, visual evaluation metrics decline steadily as difficulty increases, showing that demonstrations with more controls are harder to render accurately. *These results indicate that functional correctness is largely insensitive to component count, while visual fidelity is strongly affected by task complexity.*

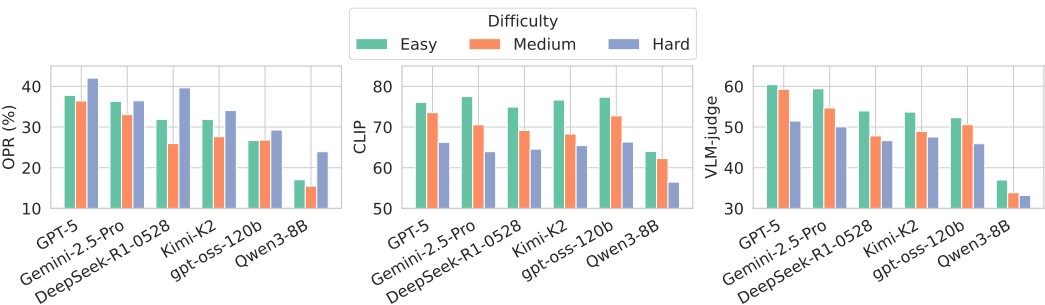

Figure 3: Performance of LLMs across different difficulty levels.

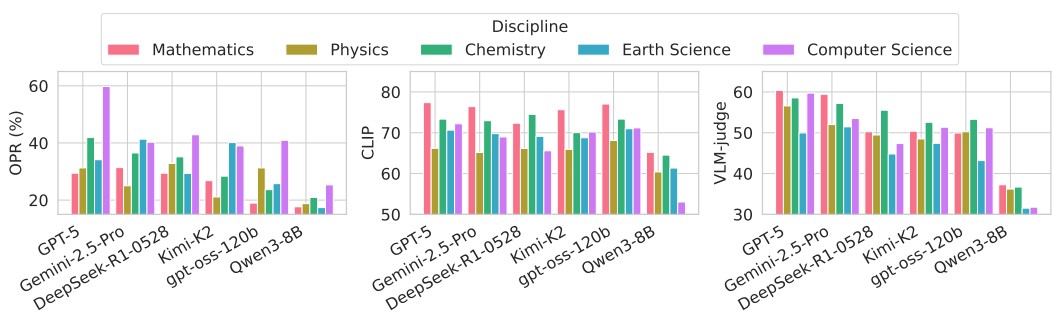

Figure 4: Performance of LLMs across different disciplines.

Figure 4 shows performance across disciplines. Models achieve the higher PFT scores on disciplines with simpler or more uniform control types, such as Computer Science and Chemistry, which typically involve only basic controls like click buttons. In contrast, VQT scores are lower for disciplines with complex visualizations, such as Physics and Earth Science, where demonstrations often require animations or 3D effects, and higher for Mathematics, which typically involves simpler, static visuals. *These patterns indicate that different disciplines place distinct demands on model capabilities, with some emphasizing accurate control logic and others requiring sophisticated visual rendering.*

**Results on Multimodal LLMs with Reference Snapshots as Input.** To evaluate the impact of reference visual context, we test several multimodal LLMs by providing them with varying numbers of reference snapshots as part of the input. The models include GPT-5, GPT-4o, Gemini-2.5-Pro, and Qwen2.5-VL-72B-Instruct. As shown in Figure 5, adding reference snapshots generally provides modest improvements across all metrics. For example, GPT-5's PFT slightly increases from 39.47% to 42.53% and CLIP scores improve from 71.95 to 73.51 as more images are added. VLM-judge scores show a similar trend, though fluctuations are observed depending on the model and number of images. Notably, some models, such as Qwen2.5-VL-72B-Instruct, experience occasional drops in certain metrics when additional snapshots are included, suggesting that extra visual input can sometimes introduce confusion rather than aid reasoning. *These results indicate that reference snapshots can enhance multimodal understanding and visual fidelity, but the effect is model-dependent and not uniformly beneficial.*

Table 3: Spearman correlation between VLM-Judge scores and human expert scores under different input configurations.

| Judge Input | Corr. |
|---|---|
| $I_{\text{gen}} + I_{\text{ref}} + L$ | **0.8827** |
| $I_{\text{gen}} + L$ | 0.8224 |
| $I_{\text{gen}} + I_{\text{ref}}$ | 0.3837 |
| $I_{\text{gen}}$ | 0.7360 |
| $C$ | 0.1408 |

**Comparison of VLM-as-Judge Configurations.** To validate our VLM-as-judge design, we measure how well different input configurations align with human judgment. Human experts score 30 randomly sampled outputs to establish ground truth, and we then conduct an ablation study by eval-

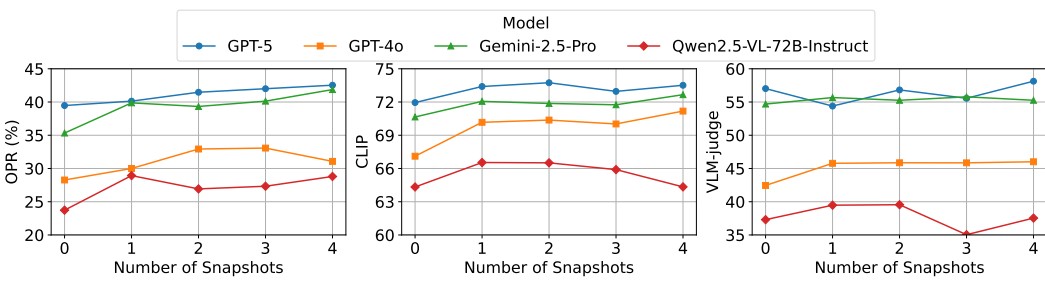

Figure 5: Performance of multimodal LLMs under varying numbers of reference snapshot inputs.

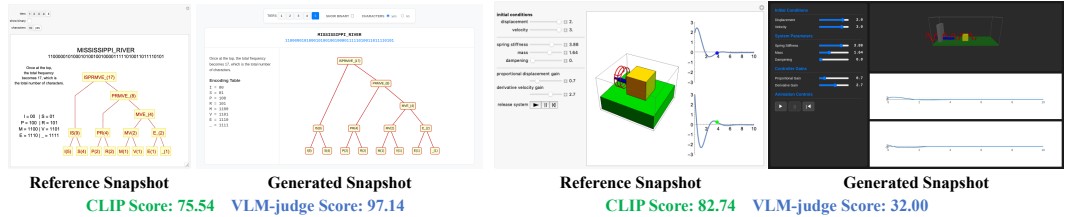

(a) Reference and generated snapshots for a Huffman Tree Encoding demonstration.

(b) Reference and generated snapshots for a Spring-Mass-Damper System demonstration.

Figure 6: Example snapshots that illustrate the complementarity of CLIP and VLM-judge scores.

uating the same outputs with configurations that remove the reference snapshot ($I_{\mathrm{ref}}$) or the checklist ($L$). We employ Spearman correlation between each configuration's scores and the human scores assesses alignment. The results in Table 3 show that the full configuration achieves the strongest alignment, highlighting the importance of both reference and checklist. Removing the checklist drops correlation to 0.3837, suggesting that without explicit checkpoints the VLM relies on coarse visual similarity, favoring outputs that appear plausible but fail scientifically. Judging with only the textual code ($C$) yields negligible correlation, confirming that visual input is essential.

**Complementarity of CLIP and VLM Judges.** Figure 6 illustrates how CLIP and VLM judges provide complementary perspectives on visual evaluation. In Figure 6a, a generated *Huffman Tree Encoding* demonstration receives a CLIP score of 75.54 and a VLM-judge score of 97.14, showing that both visual similarity and semantic correctness are well preserved. By contrast, Figure 6b presents a generated *Spring-Mass-Damper System* demonstration with a high CLIP score of 82.74 but a low VLM-judge score of 32.00. While the overall layout and graphical style are maintained, the 3D spring is rendered incorrectly, resulting in a clear scientific error. These examples show that perceptual similarity alone cannot guarantee correctness, and VLM judges are crucial for identifying semantic inaccuracies. Additional snapshots from different models are included in Appendix D.

# 6 CONCLUSION

In this work, we address the challenge of evaluating the ability of LLMs to integrate scientific knowledge into interactive demonstrations. We introduce INTERACTSCIENCE, a novel benchmark for scientific demonstration code generation with a hybrid evaluation framework combines deterministic PFT for verifying interactive functionality and reliable VQT for assessing visual fidelity.

While our study demonstrates the feasibility of evaluating scientific demonstration code generation, the current dataset is relatively limited in data size and expert verification, which may leave some subtle interactive or semantic cases untested. These aspects suggest opportunities for future work, including expanding the benchmark, incorporating broader expert validation, and exploring agent-driven testing to improve coverage, flexibility, and scalability. A more detailed discussion of these points is provided in Appendix F. We hope that our work contributes to the development of more reliable AI tools for science and education applications.

## ETHICS STATEMENT

This work uses publicly available scientific demonstrations and reference snapshots from the Wolfram Demonstrations Project under its CC BY-NC-SA license. The data are used solely for research in constructing and evaluating the INTERACTSCIENCE benchmark, and contain only implementation plans and reference snapshots without personal or sensitive information. All processing and annotation steps follow legal and ethical standards. Demonstrations were selected to ensure accurate representation of their scientific content.

We use LLMs to aid or polish writing and details are in Appendix G.

## REPRODUCIBILITY STATEMENT

We ensure reproducibility by providing all critical implementation details, including the benchmark construction process (Section 4.2) and the configuration of our experimental evaluations (Section 5.1). Comprehensive prompt for evaluation suite synthesis, including implementation plans, test cases, unit test scripts, and evaluation checklists, are given in Appendix E. All code, evaluation suites, and outputs are provided in Supplementary Materials. We will publicly release the full resources after the review process to facilitate reuse and further research.

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

## A EXTENDED RELATED WORK

**Benchmarks for Code Generation.** Existing benchmarks for code generation have largely focused on either algorithmic tasks or data visualization. Classic datasets such as HumanEval (Chen et al., 2021), MBPP (Austin et al., 2021), and their recent extensions LiveCodeBench (Jain et al., 2025) and BigCodeBench (Zhuo et al., 2024) evaluate models through hidden unit tests on programming or competitive coding problems, capturing algorithmic correctness but ignoring interactivity and visual fidelity. In parallel, visualization-oriented benchmarks such as VisCoder (Ni et al., 2025), ChartCoder (Zhao et al., 2025), DrawingPandas (Galimzyanov et al., 2025), MatPlotAgent (Yang et al., 2024), ChartMimic (Yang et al., 2025b), and CoSyn (Yang et al., 2025c) target chart generation and figure reproduction, advancing semantic and visual evaluation but remaining limited to static graphics. None of these efforts assess event-driven functional correctness or the ability to synthesize interactive scientific demonstrations. Our INTERACTSCIENCE benchmark addresses this gap by combining deterministic functional testing with visually grounded evaluation, enabling rigorous assessment of interactive code generation.

## B QUALITY VERIFICATION OF SYNTHESIZED EVALUATION SUITE

To verify the quality of the synthesized evaluation suite in the early construction stage, we randomly sampled 10 synthesized instances and manually rated each component on a 1–5 scale (higher is

Table 4: Manual rating results on 10 sampled synthesized instances.

| Component | Faithfulness | Correctness |
|---|---|---|
| Implementation Plans | 4.5 | 4.8 |
| PFT Test Cases | 4.4 | 4.7 |
| VQT Test Cases | 4.5 | 4.7 |
| PFT Unit Test Scripts | 4.6 | 4.1 |
| VQT Unit Test Scripts | 4.7 | 4.2 |
| Checklists | 4.9 | 4.7 |

better). Each rating considers two aspects, the faithfulness to the original demonstration and the correctness of the content. The average scores are reported in Table 4. Overall, all components achieve average scores above 4, with checklists rated the highest, implementation plans and test cases performing consistently well, and unit test scripts receiving relatively lower scores. This is mainly because generating executable test code is more challenging than generating descriptive natural language text, but the results still indicate that the synthesized evaluation suite is relatively reliable.

## C  EVALUATION DETAILS

### C.1  EVALUATION ENVIRONMENT

All testing after obtaining model outputs was conducted on a single server node equipped with 64 CPU cores and 512 GB of RAM. For the experiments, closed-source models and open-source models with more than 72B parameters were evaluated via standard API calls with default configuration. Open-source models with fewer than 72B parameters were deployed and run on a setup of 8 NVIDIA H800 GPUs, each with 80 GB of memory. During inference, the temperature was set to 0, and the maximum context length for open-source models was set to 32,000 tokens.

### C.2  EVALUATION COST

We report the computational and financial cost of evaluating INTERACTSCIENCE. The benchmark comprises 779 PFT unit tests and 590 VQT unit tests. Using 96 concurrent processes, the average runtime per problem is 4.15 minutes for PFT and 3.56 minutes for VQT. In worst cases, due to code errors triggering repeated timeouts, evaluation can take up to around 10 minutes. Semantic correctness in VQT is assessed using Gemini-2.5-Pro as the VLM-as-Judge. Each evaluation round involves 597 judgment queries, incurring an average cost of 8–10 USD. This demonstrates that our evaluation is practical and economically friendly, making the benchmark accessible for future research and reuse.

### C.3  METRICS COMPUTATION

For VQT, the VLM-as-Judge assigns a score on a 1–5 scale for each snapshot, reflecting the degree of semantic and scientific correctness. If the corresponding input state fails to execute all specified actions and thus produces no snapshot, the case is assigned a score of 0. Consequently, the effective scoring range becomes 0–5. For presentation clarity, we linearly rescale these scores to a 0–100 range in the reported tables and figures.

## D  MODEL OUTPUT SAMPLES

Figures 7 and 8 present the reference snapshots alongside the outputs rendered by GPT-5, Gemini-2.5-Pro, DeepSeek-R1-0528, and Qwen3-8B for two benchmark demonstrations, *Fields of Magnet Array* and *Interwoven Spherical Triangles*. Each visualization is accompanied by its CLIP and VLM-judge scores. These examples show that for complex demonstrations, different models exhibit varying levels of fidelity in both functional rendering and scientific visualization, with the quantitative scores reflecting the perceived visual quality and correctness.

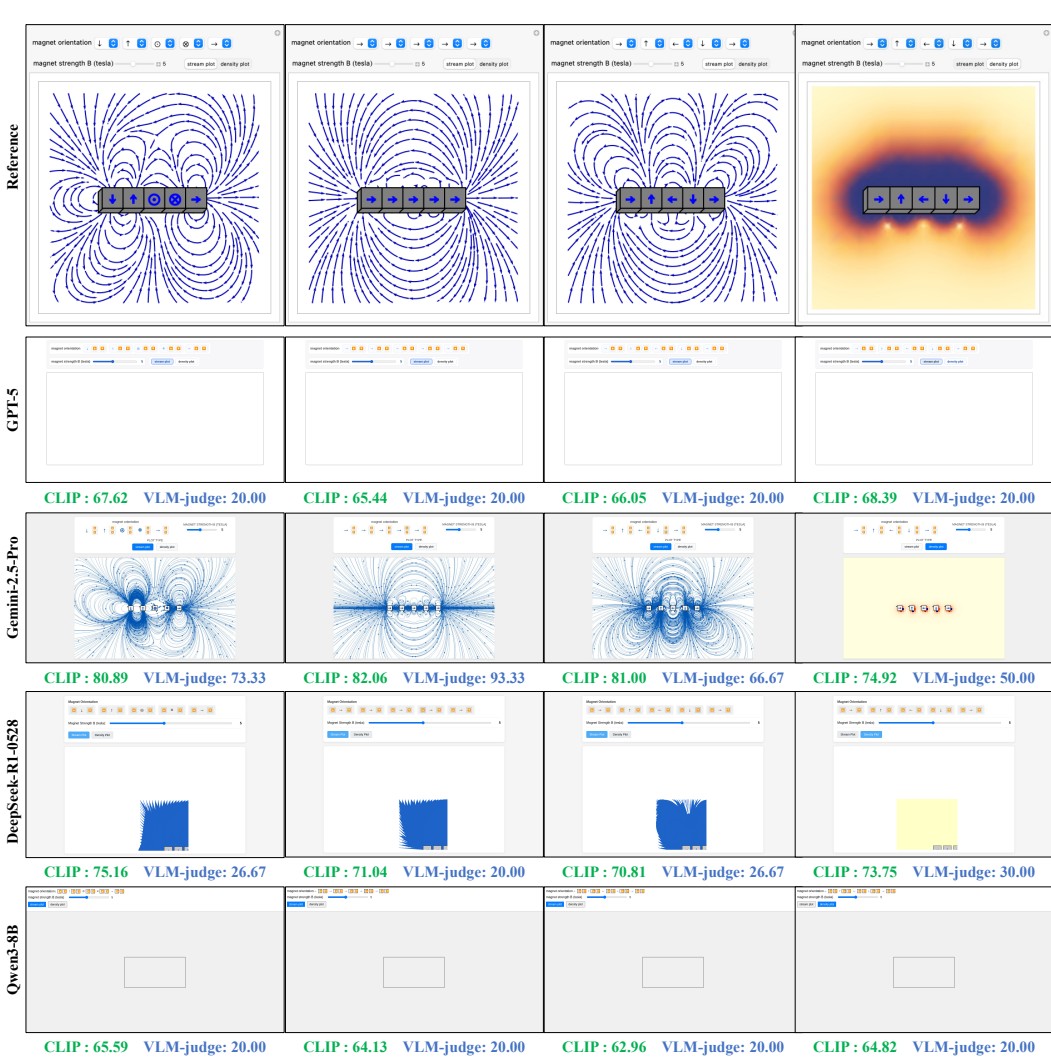

Figure 7: Reference and generated snapshots of different models for a Fields of Magnet Array demonstration.

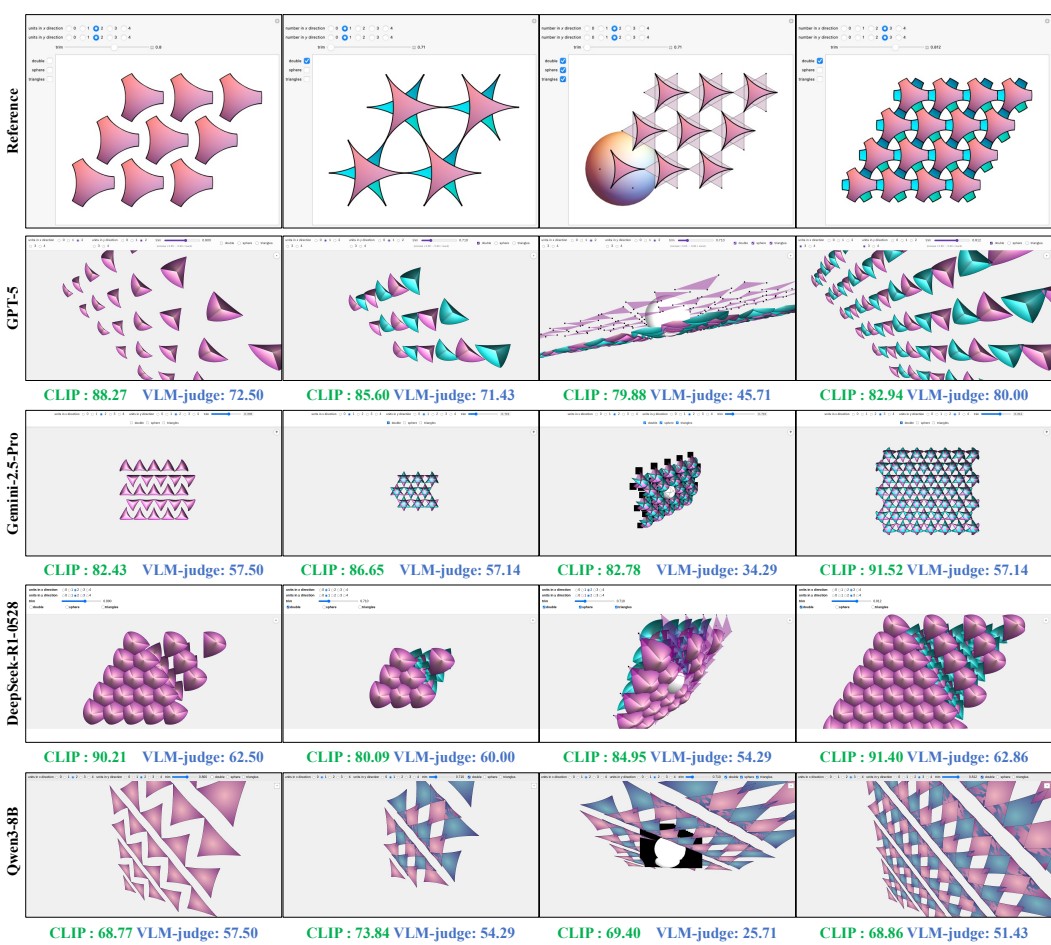

Figure 8: Reference and generated snapshots of different models for a Interwoven Spherical Triangles demonstration.

# E    PROMPTS

We provide the full system prompts used to synthesize the evaluation suites of INTERACTSCIENCE. These prompts guide Gemini-2.5-Pro in generating implementation plans (Figure 9), PFT test cases (Figure 10), VQT test cases (Figure 11), PFT unit test scripts (Figure 12), VQT unit test scripts (Figure 13), VQT checklists (Figures 14, 15, and 16), and VLM judgments (Figure 17), ensuring consistency and reproducibility of the benchmark.

# F    DISCUSSION

## F.1    LIMITATIONS.

Our current test suites are primarily synthesized using Gemini-2.5-Pro. Due to constraints in both domain expertise and annotation costs, we were only able to recruit a small group of graduate students in computer science to validate the generated test scripts and checklists. Their verification combined manual inspection with rule-based checks, allowing us to identify and fix major errors. However, this process cannot guarantee that the entire test suite is free of subtle flaws or omissions, and further large-scale expert validation remains an open challenge.

## F.2    FURTHER WORK.

For further work, our evaluation framework is not limited to scientific demonstrations. Given the ease of collecting snapshots, it can be naturally extended to the evaluation of general interactive web applications (Zhang et al., 2025; Chen et al., 2025). Furthermore, with the rapid progress of GUI-based agents (Qin et al., 2025; Sun et al., 2025b; Liu et al., 2025), agent-driven testing represents a promising future direction that could offer more flexibility and broader applicability than the current scripted approach (Wang et al., 2025; Liang et al., 2024).

# G    LLM USAGE IN WRITING

We used LLMs to aid and polish the writing of this paper, as well as to assist in the interpretation of experimental results. The models were not involved in research ideation, experimental design, or implementation. Full responsibility for the content remains with the authors.

**System Prompt - Implementation Plan Synthesis**

You are an expert in frontend web development and scientific visualization. You are given the title, description, topics, and one or more screenshots of an interactive scientific demo. This demo is designed to explain a mathematical or scientific theorem or concept through visual interaction.

Your task is to generate a precise **implementation plan** for an interactive scientific demo that explains a specific mathematical or scientific theorem. Based on the input, produce a **complete, structured, and technically feasible implementation plan** using only standard web technologies.

This plan will be used as input for a large language model to reproduce the original demo. Therefore, you must provide **exhaustive specifications** for every component: layout, content, component IDs, initial values, interactions, and visual logic.

### Constraints:
1. The demo must be implemented as a single standalone HTML file with inline HTML, CSS, and JavaScript.
2. External libraries may only be included via **CDN** (e.g., p5.js, three.js, D3.js, Plotly.js, MathJax).
3. Do **not** reference any external or uploaded media assets (images, videos, audio), and do **not** use base64-encoded binaries.
4. The plan **must fully describe the observed UI state and behavior** in the provided screenshots (including default values, rendered formulas, slider settings, etc.).
5. Require the large language model that uses this plan to **strictly follow the implementation instructions**—any missing information will lead to incorrect results.

### Output Format (strictly follow this structure, no extra commentary or code):
1. Page Content Structure
Describe each logical UI section (e.g., Title, Description, Control Panel, Graph Area, Formula Display) and its role.
2. HTML Components
List **all required HTML elements**, grouped by section. Include their types (e.g., `<div>`, `<input type="range">`, `<canvas>`, `<button>`, `<select>`).
Note if MathJax is required for formula rendering.
3. Component IDs and State
For every interactive component (sliders, checkboxes, dropdowns, buttons, etc.):
- Assign a unique `id` (e.g., `slider-angle`, `btn-play`)
- Provide: Initial/default value; Minimum and maximum (and step, if applicable); Label text or tooltip, if any
4. Interaction Logic
Explain **exactly** how each control affects the interface. For each user interaction, describe:
- What changes in the visual (e.g., redraws, updates)
- What dependent values or formulas update
- Whether animation or resets are triggered
Do not omit any interaction shown in the screenshots.
5. Visualization Techniques
Specify the rendering strategy and technology for each visual element:
- p5.js or Canvas API for custom 2D graphics
- three.js for 3D scenes
- D3.js or SVG for dynamic diagrams
- Plotly.js for charts or plots
- leaflet.js for maps
- MathJax for math formula rendering
- CSS for styling and layout (e.g., flex/grid, transitions, color indicators)
Indicate which elements require real-time updates or animation.

The resulting plan must be detailed enough that a large language model can accurately reproduce the entire original demo, including all interactions and visuals.

Here is the imformation of this scientific demo:
Name: { Name }
Description: { Description }
Topics: { Topics }
Snapshots: { Snapshots }

Figure 9: System prompt for implementation plan synthesis.

**System Prompt - PFT Test Case Synthesis**

You are an expert in frontend web development and scientific visualization. You are given the title, description, topics, one or more screenshots, and a detailed HTML implementation plan of an interactive scientific demo. Your task is to generate **component-level test cases** for each interactive element described in the plan.

Each test case should correspond to **exactly one component**, such as a slider, button, checkbox, dropdown, or any user-manipulable control.

### Test Case Requirements:
- The test case must validate:
1. The component is visible on page load.
2. The component has the correct default value or state (as defined in the plan).
3. The component can be interacted with correctly (e.g., drag slider, click button).
4. Boundary behavior should be tested (e.g., min/max values, reset, toggle on/off).
5. The interaction causes some change to the diagram, equation, UI element, or output (verify change occurred, not correctness).

### Output Format:
For each component, write one test case in the following format:
- Title: [Short description of the control being tested]
- Steps & Assertions:
1. Assert: [Component is visible]
2. Assert: [Component has correct default value or state]
3. Action: [Perform a realistic user interaction]
4. Assert: [UI update or state change occurred]
5. Action: [Boundary interaction or reset]
6. Assert: [System handles boundary or reset with some change]

### Guidelines:
- The difficulty of each test case should be **moderate**—not overly simple, not overly complex.
- Do **not** invent behavior not described in the implementation plan.
- Use only what is described or visible in the plan and screenshots.
- One case per component.
- Focus on detecting **change** rather than validating scientific correctness.
- Keep your language concise and precise—do not add explanations or commentary.

Here is the imformation of this scientific demo and the corresponding implementation plan:
Name: { Name }
Description: { Description }
Topics: { Topics }
Implementation plan:{Implementation plan}
Snapshots: { Snapshots }

Figure 10: System prompt for PFT test case synthesis.

**System Prompt - VQT Test Case Synthesis**

You are an expert in frontend web development and scientific visualization. You are given the title, description, topics, one or more screenshots, and a detailed HTML implementation plan of an interactive scientific demo. Your task is to generate **snapshot-level test cases** that replicate each screenshot through a sequence of realistic UI actions.

### Each test case should:
- Correspond to **one screenshot in the order they appear**.
- Use only **user actions** to reproduce the final state.
- Finish with a screenshot capture of the UI for comparison.

### Test Case Format:
- Title: [Short description of the visual state in screenshot 1/2/3/4]
- Steps:
1. Action: [Simulate the first interaction to reach this state]
2. Action: [Next interaction, if any]
...
N. Action: [Final interaction needed to match the screenshot]
N+1. Assert: Take a screenshot of the current UI state

### Guidelines:
- All interactions must be **derived strictly from the screenshot and the design plan**.
- Test cases must follow the **exact order of input screenshots** (first test case for first screenshot, second test case for second screenshot, etc.).
- If a specific input value is visible (e.g., slider at 0.8), use it.
- If no exact value is visible, describe the interaction in **precise relative terms** (e.g., "drag to 70% of the bar", "click second radio button from left").
- Do **not** speculate about unseen or undocumented behavior.
- The difficulty of each case should be **appropriate**: not trivial (e.g., only opening a page), and not too complex (e.g., involving inferred logic beyond the plan).
- The goal is **accurate UI state replication**, not internal logic testing.

Here is the imformation of this scientific demo and the corresponding implementation plan:
Name: { Name }
Description: { Description }
Topics: { Topics }
Implementation plan:{Implementation plan}
Snapshots: { Snapshots }

Figure 11: System prompt for VQT test case synthesis.

**System Prompt - PFT Unit Test Script Synthesis**

You are an expert in frontend web development (HTML, JavaScript, CSS) and scientific visualization. Your task is to generate **Playwright test code** for an interactive HTML page, based on a provided implementation plan and a set of structured **snapshot-level test cases**.

The input includes:
- A detailed **HTML implementation plan**, describing the layout, interactive controls, and how UI state changes based on user input.
- A set of **snapshot-level test cases**, where each test case corresponds to one screenshot and specifies a sequence of user actions to reproduce that state.

### Requirements:
Generate valid Playwright `.spec.js` test code that:
1. Navigates to the local HTML page using the code below.
2. Performs **only the exact user actions** listed in each test case (e.g., drag, click, input).
3. Uses the DOM structure and component IDs specified in the plan—do not guess or infer selectors beyond what is provided.
4. After executing all actions, takes a full-page screenshot of the resulting UI state and saves it as:
`await page.screenshot({{path:'./snapshots/{id}-[i].png',fullPage:true}});`
where [i] is the index of the test case starting from 1.
5. Ensures that tests are strictly based on the input plan and test cases—do not invent new behaviors or UI logic.
6. Does not rely on function readiness unless stated—only perform initial navigation and DOM load.

### Test Setup:
Load the HTML file using:
`const fileUrl='file://'+require('path').resolve(__dirname,'../pages/{id}.html');}`
No need to wait for external scripts or function readiness beyond page load.

### Output format:
- You must generate only valid complete Playwright test code in JavaScript wrapped in
` ```javascript ` and ` ``` ` without any explanation.
- Group test cases using `test.describe()` (per group if available).
- Define each test using `test()` with the given test case title.

Here is the implementation plan and test cases:
Implementation plan:{Implementation plan}
Test cases:{Test cases}

Figure 12: System prompt for PFT unit test script synthesis.

**System Prompt - VQT Unit Test Script Synthesis**

You are an expert in frontend web development (HTML, JavaScript, CSS) and scientific visualization. Your task is to generate **Playwright test code** for an interactive HTML page, based on a provided implementation plan and a set of structured **snapshot-level test cases**.

The input includes:
- A detailed **HTML implementation plan**, describing the layout, interactive controls, and how UI state changes based on user input.
- A set of **snapshot-level test cases**, where each test case corresponds to one screenshot and specifies a sequence of user actions to reproduce that state.

### Requirements:
Generate valid Playwright `.spec.js` test code that:
1. Navigates to the local HTML page using the code below.
2. Performs **only the exact user actions** listed in each test case (e.g., drag, click, input).
3. Uses the DOM structure and component IDs specified in the plan—do not guess or infer selectors beyond what is provided.
4. After executing all actions, takes a full-page screenshot of the resulting UI state and saves it as:
`await page.screenshot({{path:'./snapshots/{id}-[i].png',fullPage:true}});`
where [i] is the index of the test case starting from 1.
5. Ensures that tests are strictly based on the input plan and test cases—do not invent new behaviors or UI logic.
6. Does not rely on function readiness unless stated—only perform initial navigation and DOM load.

### Test Setup:
Load the HTML file using:
`const fileUrl='file://'+require('path').resolve(__dirname,'../pages/{id}.html');`
No need to wait for external scripts or function readiness beyond page load.

### Output format:
- You must generate only valid complete Playwright test code in JavaScript wrapped in
` ```javascript ` and ` ``` ` without any explanation.
- Group test cases using `test.describe()` (per group if available).
- Define each test using `test()` with the given test case title.

Here is the implementation plan and test cases:
Implementation plan:{Implementation plan}
Test cases:{Test cases}

Figure 13: System prompt for VQT unit test script synthesis.

**System Prompt - VQT Checklist Synthesis**

You are an expert in frontend web development (HTML, JavaScript, CSS) and scientific visualization.

You are given:
1. A **detailed implementation plan** of an interactive scientific demo, describing the UI structure, control elements (sliders, buttons, dropdowns, text fields), and the theorem or scientific principle the demo explains.
2. A set of **screenshots of the demo under different input states**. Each screenshot shows:
* The **input snapshot** (current state of controls such as slider values, button toggles, dropdown selections).
* The **visual output** (graph, diagram, simulation, or formula rendering) produced by the demo under that input.

Your task is to generate a **checklist for each screenshot**, which specifies the scientific correctness criteria of the **visual output** given that input state.

### Requirements for the checklist
1. **Checklist is output-oriented**
* Do not check whether buttons, sliders, or controls are styled correctly.
* Treat control states only as **inputs** that determine what the visualization should show.
* Focus all checklist items on verifying whether the **visual output image** is scientifically correct.
2. **Connect input to output explicitly**
* Every checklist item must link the given **input state** to the expected **output visualization**.
* Example: *If the angle slider is set to 45°, then the plotted projectile trajectory must peak at the midpoint of its range.*
3. **Scientific focus**
* For math demos: formulas, curves, intersections, asymptotes.
* For physics demos: motion paths, conservation laws, force vectors.
* For geometry demos: shapes, proportions, congruency.
* For statistics/plots: distributions, scaling, labeled values.
4. **Visual verification**
* All checklist items must be verifiable by comparing the **screenshot output image** with a reference screenshot.
* Do not assume hidden internal states or unseen code behavior.
5. **Do not go beyond the plan**
* Only include checklist items that are **explicitly described in the implementation plan** *and* are **visible in the screenshot**.
* Do not invent or infer behavior that is not both in the plan and observable in the screenshot.
* If something is in the plan but not visible in the screenshot, **do not include it**.

Figure 14: System prompt for VQT checklist synthesis Part 1.

```
### Output Format (strict JSON)

{
  "screenshot_id": "n",
  "input_state": {
    "control_name_1": "value",
    "control_name_2": "value"
  },
  "checklist": [
    {
      "category": "Formula correctness",
      "expectation": "{expected formula or update, but only if
      both plan and screenshot show formula}"
    },
    {
      "category": "Graph/Diagram correctness",
      "expectation": "{expected graph or shape properties, if
      plan defines them and screenshot shows them}"
    },
    {
      "category": "Axes/Labels correctness",
      "expectation": "{expected axis labels/ranges/units, only
      if defined in plan and visible}"
    },
    {
      "category": "Numeric outputs correctness",
      "expectation": "{expected numerical outputs, but only if
      plan specifies their presence and screenshot shows them}"
    },
    {
      "category": "Consistency with input state",
      "expectation": "{visual updates must reflect given input
      controls, only if plan links them to visuals}"
    }
  ]
}
```

Figure 15: System prompt for VQT checklist synthesis Part 2.

```
### Example (from a projectile motion demo plan)
**Screenshot input state**: angle slider = 30°, initial speed = 20 m/s
{
  "screenshot_id": "1",
  "input_state": {
    "angle_slider": "30°",
    "initial_speed": "20 m/s"
  },
  "checklist": [
    {
      "category": "Formula correctness",
      "expectation": "Displayed trajectory equation includes theta =
      30° and v = 20 m/s, as defined in the plan"
    },
    {
      "category": "Graph/Diagram correctness",
      "expectation": "Trajectory curve is parabolic as specified in
      plan; arc matches input parameters"
    },
    {
      "category": "Axes/Labels correctness",
      "expectation": "Axes labeled with meters as required by plan"
    },
    {
      "category": "Numeric outputs correctness",
      "expectation": "Maximum height and range values are displayed
      if defined in plan and visible in screenshot"
    },
    {
      "category": "Consistency with input state",
      "expectation": "Visualization reflects input slider angle = 30°"
    }
  ]
}
Here is the implementation plan and snapshots:
ID: {ID}
Implementation plan:{Implementation plan}
Snapshots:{Snapshots}
```

Figure 16: System prompt for VQT checklist synthesis Part 3.

**System Prompt - VQT VLM-as-Judge**
You are an expert judge for evaluating scientific visualization demos.

You are given:
1. A **reference screenshot** that represents the correct output of the demo under a specific input state.
2. A **generated screenshot** from a candidate implementation under the same input state.
3. A **checklist** of verification items describing what scientific properties must be visible and correct in the output image.

### Your task
1. Carefully compare the **generated screenshot** with the **reference screenshot**.
2. For each **checklist item**, assign a score from **1 to 5** using the rubric below.
3. Provide a short justification for each score.

### Scoring Rubric
* **5 (Perfect / Fully Correct)**
* Output image matches the reference screenshot precisely for this checklist item.
* No scientific or visual errors observed.
* **4 (Minor Deviation)**
* Output image mostly matches, but there are small differences (e.g., slight shift in curve, small scaling error, minor misalignment) that do not change the core scientific correctness.
* **3 (Partial Correctness)**
* Some parts are correct (e.g., correct general shape, but wrong labels; correct axis scaling, but wrong curve peak).
* Noticeable deviation from reference that may reduce scientific clarity.
* **2 (Mostly Incorrect)**
* The item is largely wrong, but a small aspect is still correct (e.g., axis present but mislabeled, trajectory drawn but incorrect path).
* **1 (Completely Incorrect / Missing)**
* The expected scientific property is entirely absent or completely wrong.
* Visualization contradicts the reference screenshot.

### Output Format (strict JSON)

```
{
  "checklist_results": [
    {
      "expectation": "{text from checklist item}",
      "score": 4,
      "reason": "Trajectory shape matches but peak position is
      slightly shifted."
    },
    {
      "expectation": "{text from checklist item}",
      "score": 5,
      "reason": "Axis labels and scaling are identical to reference."
    }
  ]
}
```

Here is the checklist, reference snapshot, and generated snapshot:
Checklist: {Checklist}
Reference snapshot:{Reference snapshot}
Generated snapshot:{Generated snapshot}

Figure 17: System prompt for VQT VLM-as-Judge.

