# OpenReview forum: "InteractScience: Programmatic and Visually-Grounded Evaluation of Interactive Scientific Demonstration Code Generation"
_ICLR.cc/2026/Conference — Submitted to ICLR 2026_

### Official Review · Reviewer_ei4T · 2025-10-26

**Soundness:** 3
**Presentation:** 3
**Contribution:** 2
**Rating:** 4
**Confidence:** 3

**Summary:**

This work describes a benchmark project of using large language models (LLMs) for creating scientific demos, spanning over five domains on 150 tasks. The authors employed 30 models for this task, and evaluated them by the accuracy of the content through test cases (LLM-generated) and the interactive performances (CLIP and VLM-Judge). This work shows how LLMs currently perform in the domain of scientific demo generation.

**Strengths:**

+ The work is easy to follow. Overall, the communication of the work is smooth and clear.
+ The motivation of the work is well-established. It has great potential to be a pivotal work in the domain of scientific demo generation.

**Weaknesses:**

- One major issue is the contribution of the work. While having these numbers showing how good each model is could be interesting for many model builders, it is not discussed how this work can be helpful in the future -- how would the authors position themselves in making the actual contribution towards these models? What do these results mean for the domain? These are way more important questions that the paper has a chance to discuss, but fails to do.
- Another major limitation of the work is the missed details on many implementation processes. The details are below in questions, and some of them can be quite important validation questions.

**Questions:**

- Lines 28-31: How exactly does the work help the domain despite sharing a bunch of numbers showing the model is not working very well now? This is not clear.
- Line 47: Note: This is a fairly wide range scope claimed when talking about scientific demonstration code generation.
- Line 66 Figure 1: three types of tasks. Do you have a reference for previous work categorizing tasks this way? This may (optionally) require some taxonomy prior work.
- Line 91 VLM as judge: How accurate is this? The targeted audiences are also important. For example, how students perceive this will be different from how the general public does. Student levels are also important (elementary or high school?)
- Related work: As commented, the scope of scientific visualization is too broad. Until now, we still don't know what exact domain/context we are trying to create interactive demos for. This is unclear to audiences.
- Line 249: The exact selection criteria are needed.
- Line 252: About the difficulty of the demos: How did you define the difficulty of problems/demos?
- Line 257: Did you validate these generated tests to see if they work well?
- Conclusion section: Results are shared and described in detail. However, there's no discussion on what this means in the main context.

---

> ### Author Response · Authors · 2025-11-25
>
> Dear Reviewer ei4T，
>
> We sincerely thank you for your positive assessment of the paper's clarity and motivation. We appreciate your insightful questions regarding the broader impact of our work and the specific implementation details. Below, we provide detailed responses to your concerns.
>
> #### 1. On Contribution and Future Impact (Weakness 1, Questions 1 & 9)
>
> > **Weakness 1:** One major issue is the contribution of the work. While having these numbers showing how good each model is could be interesting for many model builders, it is not discussed how this work can be helpful in the future -- how would the authors position themselves in making the actual contribution towards these models? What do these results mean for the domain? These are way more important questions that the paper has a chance to discuss, but fails to do.
> **Question 1:** Lines 28-31: How exactly does the work help the domain despite sharing a bunch of numbers showing the model is not working very well now? This is not clear.
> **Question 9:** Conclusion section: Results are shared and described in detail. However, there's no discussion on what this means in the main context.
>
> You asked how this work helps the domain beyond showing that models currently fail.
>
> **Clarification:**
>
> The primary contribution of INTERACTSCIENCE is diagnosing why models fail, which provides a roadmap for future improvement.
> - **Identifying the "Superficial Competence" Gap:** Our results reveal that models achieve **high Action Success Rates (>85%)** (generating working buttons/sliders) but **low Scientific Correctness (<55)**. This proves that current models treat scientific demos as "UI tasks" rather than "reasoning tasks."
> - **Guiding Model Development:** By decoupling Programmatic Functional Testing (PFT) from Visually-Grounded Qualitative Testing (VQT), we provide model builders with a gradient for optimization. Future work must focus not on better coding syntax (which is already good), but on **grounding reasoning in visual event logic.**
>
> We will expand the Discussion (Section 6) to explicitly state these implications: to improve, models need training data that pairs scientific principles directly with interaction logic, rather than just static code or QA pairs.

---

> ### Author Response · Authors · 2025-11-25
>
> #### 2. On Implementation Details and Validation (Weakness 2, Questions 6, 7, 8)
>
> > **Weakness 2:** Another major limitation of the work is the missed details on many implementation processes. The details are below in questions, and some of them can be quite important validation questions.
> **Question 6:** Line 249: The exact selection criteria are needed.
> **Question 7:** Line 252: About the difficulty of the demos: How did you define the difficulty of problems/demos?
> **Question 8:** Line 257: Did you validate these generated tests to see if they work well?
>
> We appreciate the opportunity to clarify the rigorous details behind our benchmark construction.
>
> - Q6: **Selection Criteria.** We selected 150 demonstrations from the Wolfram Demonstrations Project based on three criteria. **(1) Diversity:** we balanced the samples equally across five disciplines (Math, Physics, Chemistry, Earth Science, and CS) to allow domain-general conclusions. **(2) Feasibility:** we chose demonstrations that can be implemented using standard web technologies (HTML and JavaScript) rather than proprietary engines, so that both open-source and closed-source models are evaluated under the same conditions. **(3) Stability and Self-Containment:** we avoided demonstrations involving randomness or external data sources, such as those requiring online retrieval of economic statistics. This ensures that the reference snapshots remain stable gold standards and that models can produce correct outputs in an offline environment.
>
> - Q7: **Defining Difficulty.** We defined difficulty quantitatively based on the complexity of the interaction logic, specifically **the number of interactive components**:
>   - Easy: 1-3 components.
>   - Medium: 4-6 components.
>   - Hard: 7-10 components.
>   This metric correlates with the length of the implementation plan and the number of assertions required (Table 1). This metric also aligns with the VQT score trend in Figure 3, where harder questions receive lower scores. This pattern indicates that questions with more interactive components usually involve more scientific concepts and create higher implementation difficulty, which supports our choice of using component count as a quantitative difficulty measure.
>
> - Q8: **Test Validation.** Yes, we validated the synthesized tests.
>   - **Automated:** We used rule-based validation to ensure all scripts were executable Playwright code.
>   - **Human:** As detailed in Appendix B, we initially sampled instances and manually rated the synthesized test cases and plans on a 1–5 scale. Beyond this, following requests from other reviewers, we added a new validation experiment with a larger and more diverse sample. We increased the sample size to 30 questions, which is 20% of the full set of 150 questions, and ensured coverage by selecting one question from each subject and each difficulty level. The results were consistent with our earlier findings. The new results are:
>
> | Component | Faithfulness | Correctness |
> |---|---|---|
> | Implementation Plans | 4.43 | 4.73 |
> | PFT Test Cases | 4.37 | 4.63 |
> | VQT Test Cases | 4.43 | 4.67 |
> | PFT Unit Test Scripts | 4.53 | 4.27 |
> | VQT Unit Test Scripts | 4.57 | 4.23 |
> | Checklists | 4.87 | 4.63 |
>
> In addition, because unit test scripts are less readable, we used the Playwright UI Mode (https://playwright.dev/docs/test-ui-mode) to conduct an extra correctness check. For the 30 sampled questions, we tested the code generated by three models, Gemini-2.5-Pro, DeepSeek-V3, and Qwen3-8B, giving 90 unit test scripts in total. We then manually judged whether each script produced correct test behavior, that is, whether it avoided false positives and false negatives. The correctness rates (%) are:
>
> | Model | PFT Unit Test Scripts Correctness (%) | VQT Unit Test Scripts Correctness (%) |
> |---|---|---|
> | Gemini-2.5-Pro | 87.5 | 91.2 |
> | DeepSeek-V3 | 86.8 | 92.4 |
> | Qwen3-8B | 95.3 | 96.7 |
>
> These results show that almost all correctness rates exceed 85%. The VQT unit test scripts show higher correctness than PFT unit test scripts, as they do not include assertions and therefore are less likely to produce false outcomes. For code with lower implementation quality, the correctness rates exceed 95%, since the functions implemented are limited and most tests fail as expected. We will include these details and sample screenshots of the UI Mode workflow in the final version.

---

> ### Author Response · Authors · 2025-11-25
>
> #### 3. On VLM-as-Judge Accuracy and Audience (Question 4)
>
> > **Question 4:** Line 91 VLM as judge: How accurate is this? The targeted audiences are also important. For example, how students perceive this will be different from how the general public does. Student levels are also important (elementary or high school?)
>
> - **Accuracy:** We rigorously validated our VLM judge. In **Table 3**, we show that our configuration (Generated Snapshot + Reference + Checklist) achieves a **Spearman correlation of 0.8827** with human expert judges. This is a very high level of agreement.
>
> - **Target Audience:** You raised an important point about the influence of different audiences, such as students at different levels or the general public. We agree that audience-specific perception is a meaningful dimension and is worth considering in future evaluation design. In our current study, however, the judge focuses on **scientific correctness**, such as whether a trajectory is parabolic or whether energy is conserved. These elements are objective and are defined by the **Checklists**, so the "audience" for the judge is the **scientific fact** itself rather than a particular user group. We will include a short discussion in the future work section to highlight how audience-specific evaluation could be integrated into later extensions of our benchmark.
>
> #### 4. On Scope and Taxonomy (Questions 2, 3, 5)
>
> > **Question 2:** Line 47: Note: This is a fairly wide range scope claimed when talking about scientific demonstration code generation.
> **Question 3:** Line 66 Figure 1: three types of tasks. Do you have a reference for previous work categorizing tasks this way? This may (optionally) require some taxonomy prior work.
> **Questions 5:** Related work: As commented, the scope of scientific visualization is too broad. Until now, we still don't know what exact domain/context we are trying to create interactive demos for. This is unclear to audiences.
>
> - Q2 & Q5: **Scope.** We agree "Scientific Visualization" is broad. We explicitly narrow our scope to "Scientific Demonstration Code Generation" in Section 3.1: generating self-contained, interactive web applications. We will clarify in the introduction that we are not targeting static plotting (like Matplotlib) or high-performance rendering (like ParaView), but specifically web-based scientific/educational demos (code in HTML/CSS/JS). We will explain this part more clearly in the new version.
>
> - Q3: **Taxonomy (Figure 1).** The categorization in Figure 1 (Knowledge QA vs. Static Webpage vs. Interactive Demo) is a conceptual framework we introduced to position our task. It illustrates that our task is the intersection of scientific knowledge (QA) and functional coding (Webpage Gen). We will clarify that this is our proposed categorization to highlight the unique multimodal requirements of this domain.
>
> We hope these responses clarify the method's soundness and the paper's contribution. We will incorporate these details into the final version to ensure the work's impact is clearly communicated.

---

### Official Review · Reviewer_kGTP · 2025-10-27

**Soundness:** 2
**Presentation:** 2
**Contribution:** 3
**Rating:** 4
**Confidence:** 4

**Summary:**

This paper presents INTERACTSCIENCE, a new benchmark for evaluating LLMs on the task of scientific demonstration code generation. In this setting, the input is an implementation plan, while the output is a self-contained HTML file that renders an interactive scientific demonstration. The paper collects 150 examples from the Wolfram Demonstrations Project across five domains. For evaluation on these 150 examples, the authors propose Programmatic Functional Testing (PFT) -- a unit test for interaction logic -- and Visually-Grounded Qualitative Testing (VQT) -- a visual comparison using CLIP and a VLM-as-Judge. They evaluate 30 LLMs and provide quantitative results and analyses across difficulty levels and disciplines.

**Strengths:**

- S1. The paper explores a useful domain -- scientific demonstration synthesis -- that goes beyond static visualization or plain code generation. This domain is particularly relevant for education, research demonstrations, etc.
* S2. The authors introduce a systematic evaluation method combining unit tests, visual similarity measures, and checklist-based assessments. This evaluation suite can comprehensively assess the implementation quality of interactive demonstrations.
* S3. The benchmark covers multiple disciplines and difficulty levels and provides a comprehensive comparison across many models.

**Weaknesses:**

* W1. Limited novelty in the evaluation framework. The idea of designing unit tests and using a VLM-as-a-judge is not particularly new. While the framework is practical, it lacks strong originality.
- W2. VLM-as-Judge reliability remains questionable. The "Comparison of VLM-as-Judge Configurations" (lines 429–462) only investigates which input combination yields the highest correlation with human scores. This justifies the prompt design but does not establish how reliable the VLM judge is compared to human evaluation. Therefore, the overall reliability of this automatic judging process remains unaddressed. I suggest that the authors also report the reliability of the VLM-as-Judge compared with human judgments.
- W3. Potential evaluation issue. The visual evaluation assumes a single "correct" snapshot. However, many scientific demonstrations can have multiple equally valid visualizations (e.g., different color schemes, scaling, or styles). Such variability could lead to unfair penalization of valid outputs when using CLIP- or VLM-based scoring. How does the evaluation framework address this issue, or is this an inherent limitation of the proposed evaluation method?
* W4. Lack of failure analysis. The analyses focus mainly on validating design choices (e.g., whether to include reference snapshots or checklists) rather than uncovering deeper insights or failure patterns of existing models. No detailed failure analysis is provided to explain why models fail, what error types dominate, or what insights could guide domain practitioners, which weakens the paper's analytical depth.
* W5. Writing clarity and notation. Section 3.3 is hard to follow. Many notations (e.g., (t_{vqt}=(A,i_{ref},L))) are introduced only once and never reused, adding cognitive load without improving clarity. Simplifying descriptions and adding concrete examples would make the evaluation process easier to understand.

**Questions:**

See Weaknesses

---

> ### Author Response · Authors · 2025-11-25
>
> Dear Reviewer kGTP,
>
> We thank you for your high-confidence review and for recognizing the relevance of the INTERACTSCIENCE domain (S1) and the systematic nature of our evaluation suite (S2). We appreciate your constructive feedback regarding the evaluation nuances and clarity. Below, we address your concerns point-by-point.
>
> #### 1. On Novelty of the Evaluation Framework (W1)
>
> > **W1.** Limited novelty in the evaluation framework. The idea of designing unit tests and using a VLM-as-a-judge is not particularly new. While the framework is practical, it lacks strong originality.
>
> While we agree that unit testing and VLM-based judging are established techniques individually, our core contribution lies in creating a **unified evaluation method for the unique, unserved challenge of interactive scientific code**, a setting that has not been supported before.
>
> **The Gap:**
>
> Existing benchmarks generally fall into two categories, neither of which suffices for this task:
> - **Static Visualization Benchmarks:** Benchmarks like SridBench or EduVisBench focus on static images, lacking mechanisms to verify if a user's action triggers the correct response.
> - **General Code Benchmarks:** Benchmarks like Interaction2Code or WebGen-Bench focus on generic web elements. They often rely on fixed-interval screenshots or element-existence checks without verifying the complex event-driven logic required for scientific simulations.
>
> **The Hybrid Necessity:**
>
> As illustrated in Figure 1, a model can generate a functionally "correct" generic webpage (e.g., working buttons and layout) but fail the specific scientific logic (e.g., "Wrong animation direction" or "Incorrect force arrow orientation").
> - Standard functional tests cannot catch that the physics formula is wrong.
> - Standard visual metrics (like CLIP) cannot verify that a specific slider movement caused the correct dynamic update.
>
> **Integration and Uniqueness:**
>
> **Our novelty is the hybrid framework.**
> - Programmatic Functional Testing (PFT): Rigorously verifies interaction logic (e.g., "Does the slider update the variable?").
> - Visually-Grounded Qualitative Testing (VQT): Uses reference-guided checklists to verify scientific fidelity (e.g., "Is the trajectory parabolic?").
>
> To explicitly demonstrate this novelty, we provide the following comparison table (Table R1), which highlights that INTERACTSCIENCE is the first benchmark to require the intersection of **Scientific Reasoning, Interaction, and Visualization.**
>
> | Benchmark | Primary Task | Scientific Reasoning | Interaction-based Testing | Visualization-based Testing | Evaluation |
> |---|---|---|---|---|---|
> | **Scientific Visualization** |  |  |
> | SridBench | Scientific Visualization Generation | ✓ | ✗ | ✓ | LLM-judge |
> | EduVisBench | Scientific Visualization Generation | ✓ | ✗ | ✓ | LLM-judge |
> | **Software Issues Resolving** |  |  |
> | SWE-bench | Repository-level Bug Fixing | ✗ | ✓ | ✗ | Automation |
> | SWE-bench Multimodal | Repository-level Bug Fixing with Visual Context | ✗ | ✓ | ✗ | Automation |
> | **Code Generation** |  |  |
> | LiveCodeBench | Competitive-Programming Code Generation | ✗ | ✗ | ✗ | Automation |
> | Interaction2Code | Interactive Web Page Generation | ✗ | ✗ | ✗ | Automation |
> | WebGen-Bench | Interactive Web Page Generation | ✗ | ✗ | ✗ | LLM-judge |
> | WebDev Arena | Web Design | ✗ | ✗ | ✓ | Human Voting |
> | ArtifactsBench | Interactive Visual Artifacts | ✗ | ✗ | ✓ | LLM-judge |
> | InteractScience | Interactive Scientific Demonstration Generation | ✓ | ✓ | ✓ | Automation + LLM-judge |
>
> Note that INTERACTSCIENCE is the only benchmark that simultaneously evaluates scientific reasoning, interaction logic, and visual output.
> We hope this comparison clarifies that while the components of our evaluation are established, their synthesis addresses a distinct and previously open problem space.

---

> ### Author Response · Authors · 2025-11-25
>
> #### 2. On VLM-as-Judge Reliability (W2)
>
> > **W2.** VLM-as-Judge reliability remains questionable. The "Comparison of VLM-as-Judge Configurations" (lines 429–462) only investigates which input combination yields the highest correlation with human scores. This justifies the prompt design but does not establish how reliable the VLM judge is compared to human evaluation. Therefore, the overall reliability of this automatic judging process remains unaddressed. I suggest that the authors also report the reliability of the VLM-as-Judge compared with human judgments.
>
> You raised a valid concern about ensuring the reliability of the VLM judge beyond prompt optimization.
>
> As reported in **Table 3**, our proposed configuration (Generated Snapshot + Reference + Checklist) achieved a Spearman correlation of **0.8827** with human judges. In the context of LLM-as-a-Judge literature, a correlation of ~0.88 indicates very high reliability and alignment with human preference.
>
> In addition, we added experiments on the reliability of the VLM-as-Judge compared with human judgments. Recognizing that raw unit test scripts are difficult to read, we utilized Playwright UI Mode (https://playwright.dev/docs/test-ui-mode) to visually verify the execution logic. For the same 30 sampled questions, we executed the scripts generated by three distinct models (Gemini-2.5-Pro, DeepSeek-V3, and Qwen3-8B), resulting in 90 unit test executions. We manually judged whether each script produced the correct test behavior (avoiding false positives/negatives).
>
> | Model | PFT Unit Test Scripts Correctness (%) | VQT Unit Test Scripts Correctness (%) |
> |---|---|---|
> | Gemini-2.5-Pro | 87.5 | 91.2 |
> | DeepSeek-V3 | 86.8 | 92.4 |
> | Qwen3-8B | 95.3 | 96.7 |
>
> The correctness rates consistently exceed 85%. VQT scripts show higher correctness as they involve fewer logic assertions. Interestingly, for lower-capability models (like Qwen3-8B), script correctness is higher (>95%) because the generated code is simpler, making the tests behave more predictably (often failing correctly). We will include these results and sample workflow screenshots in the final version.

---

> ### Author Response · Authors · 2025-11-25
>
> #### 4. On Failure Analysis (W4)
>
> > **W4.** Lack of failure analysis. The analyses focus mainly on validating design choices (e.g., whether to include reference snapshots or checklists) rather than uncovering deeper insights or failure patterns of existing models. No detailed failure analysis is provided to explain why models fail, what error types dominate, or what insights could guide domain practitioners, which weakens the paper's analytical depth.
>
> We acknowledge that we can provide deeper qualitative insights. Our current analysis highlights a critical failure mode:
>
> - **The "Superficial Competence" Gap:** In Section 5.2, we identify a dominant failure pattern where models achieve high **Action Success Rates (>85%)**, which meaning they successfully code the UI widgets, but low **VLM scores (<55)**.
> - **Interpretation:** This indicates that models fail to bind the interactive controls to the correct underlying scientific logic. They can build the "shell" of the demo but break the "physics" inside it.
> - **Specific Example:** We illustrate this in **Figure 6**, showing a Spring-Mass-Damper system where the model generated a high-quality looking visual (CLIP score 82.74) that was scientifically wrong (VLM score 32.00) because the 3D spring rendering was incorrect.
>
> To better ground this gap in data, we sampled 30 problems and examined the outputs of three distinct models (Gemini-2.5-Pro, DeepSeek-V3, and Qwen3-8B), giving 90 cases for manual inspection. Across these samples, we classified errors into four types:  **A. Rendering Failure/Basic Error, B. Logic/Scientific Error, C. Layout Error, and D. Numerical/Parameter Error.**
> | Error Category | Gemini-2.5-Pro |  | DeepSeek-V3 |  | Qwen3-8B |  | Overall |  |
> |---|---|---|---|---|---|---|---|---|
> | A. Rendering Failure/Basic Error | 6 | 20.00% | 6 | 20.00% | 26 | 86.70% | 38 | 42.22% |
> | B. Logic/Scientific Error | 18 | 60.00% | 20 | 66.70% | 10 | 33.30% | 48 | 53.33% |
> | C. Layout Error | 14 | 46.70% | 16 | 53.30% | 5 | 16.70% | 35 | 38.89% |
> | D. Numerical/Parameter Error | 10 | 33.33% | 13 | 43.33% | 20 | 66.67 | 52 | 47.78% |
>
> In the final version, we will formalize these observations in a dedicated “Failure Analysis” subsection.
>
>
> #### 5. On Writing Clarity and Notation (W5)
>
> > **W5.** Writing clarity and notation. Section 3.3 is hard to follow. Many notations (e.g., (t_{vqt}=(A,i_{ref},L))) are introduced only once and never reused, adding cognitive load without improving clarity. Simplifying descriptions and adding concrete examples would make the evaluation process easier to understand.
>
> We accept your feedback regarding the notation in Section 3.3. We agree that notations like $t_{vqt}=(A,i_{ref},L)$ introduce unnecessary cognitive load if not reused extensively.
>
> We will simplify Section 3.3 by removing the formal notation where it is not strictly necessary for precision, replacing it with clear descriptive text and concrete examples of test cases to improve readability.
>
> We hope these clarifications address your concerns and demonstrate the robustness of the INTERACTSCIENCE benchmark.

---

### Official Review · Reviewer_df5Z · 2025-10-28

**Soundness:** 3
**Presentation:** 2
**Contribution:** 2
**Rating:** 4
**Confidence:** 3

**Summary:**

The paper introduces a comprehensive benchmark called INTERACTSCIENCE designed to evaluate large language models (LLMs)’ ability to generate interactive scientific demonstrations that combine scientific reasoning, front-end logic, and visual rendering. The study develops a hybrid evaluation framework combining Programmatic Functional Testing (PFT), which verifies user interaction logic through deterministic unit tests, and Visually-Grounded Qualitative Testing (VQT), which measures visual and semantic correctness using CLIP similarity and Vision-Language Model scoring. The resulting INTERACTSCIENCE benchmark contains 150 curated tasks from five scientific domains, each equipped with structured implementation plans, unit tests, and reference snapshots sourced from the Wolfram Demonstrations Project. Evaluations on 30 open- and closed-source models reveal that most models can produce interactive interfaces but often fail to maintain correct scientific logic, exposing a gap between functional and conceptual accuracy. Closed-source models like GPT-5 perform best overall, while large open-source models show scaling improvements. The benchmark offers the first automated, reproducible framework to jointly assess reasoning, visual fidelity, and interactivity in scientific code generation.

**Strengths:**

1. The paper defines and formalizes Scientific Demonstration Code Generation, a previously underexplored intersection of scientific reasoning, interactivity, and visualization
2. Combining deterministic testing (PFT) and perceptual/semantic evaluation (VQT) is a strong methodological innovation that goes beyond subjective visual checks.
3. The benchmark spans five domains and three difficulty levels, ensuring balanced diversity and challenging coverage.
4. Evaluating 30 models across both open and closed-source LLMs.

**Weaknesses:**

1. While correlation of VLM and human scores is measured, human involvement in data verification and semantic correctness remains minimal. The validation of synthesized test cases is important and should be included in the main content.
2. In line 259, the author mentioned “manual inspection and rule-based validation,” but did not provide sufficient details on how the inspection was conducted or what specific rules were used for validation. In Appendix B, they stated that only 10 synthesized instances were randomly sampled for manual inspection, which makes this evaluation rather limited and less convincing.
3. In figure 3, the results show better Overall Pass Rate (OPR) on hard question, which is a little bit counter-intuitive. From the analysis, the author did not provide a good explanation of this.

**Questions:**

1. How did the authors verify the correctness of the synthesized test cases?
2. For VLM-as-Judge, how were the checklists designed? How was the manual inspection conducted, and how many evaluators were involved? Additionally, how were conflicts between human evaluators resolved?

---

> ### Author Response · Authors · 2025-11-25
>
> Dear Reviewer df5Z,
>
> We sincerely thank you for your thoughtful review and for recognizing the value of INTERACTSCIENCE as a comprehensive benchmark filling an underexplored gap. We appreciate your constructive criticism regarding the data verification process and the counter-intuitive results. Below, we address your concerns with new experiment results and detailed clarifications.
>
> #### 1. On Data Verification (Weakness 1 & Question 1)
>
> > **Weakness 1:** While correlation of VLM and human scores is measured, human involvement in data verification and semantic correctness remains minimal. The validation of synthesized test cases is important and should be included in the main content.
> **Question 1:** How did the authors verify the correctness of the synthesized test cases?
>
> We acknowledge that the initial random sampling of 10 instances in Appendix B was limited. To address this and robustly verify the correctness of our synthesized test cases, we conducted a **new, expanded validation experiment**.
> We increased the sample size to **30 questions (20% of the full dataset)**, ensuring balanced coverage by selecting one question from each subject and each difficulty level. We recruited 5 graduate students as annotators to manually rate the synthesized components on a 1–5 scale. The results, consistent with our earlier findings, are presented below:
>
> | Component| Faithfulness| Correctness|
> |---|---|---|
> | Implementation Plans| 4.43| 4.73 	|
> | PFT Test Cases| 4.37| 4.63 |
> | VQT Test Cases| 4.43 	| 4.67 |
> | PFT Unit Test Scripts | 4.53 | 4.27 |
> | VQT Unit Test Scripts| 4.57 | 4.23 |
> | Checklists | 4.87 | 4.63 |
>
> Recognizing that raw unit test scripts are difficult to read, we utilized **Playwright UI Mode** (https://playwright.dev/docs/test-ui-mode) to visually verify the test execution logic. For the same 30 sampled questions, we executed the scripts generated by three distinct models (Gemini-2.5-Pro, DeepSeek-V3, and Qwen3-8B), resulting in **90 unit test executions**. We manually judged whether each script produced the correct test behavior (avoiding false positives/negatives).
>
> | Model | PFT Unit Test Scripts Correctness (%) | VQT Unit Test Scripts Correctness (%) |
> |---|---|---|
> | Gemini-2.5-Pro | 87.5 | 91.2|
> | DeepSeek-V3 | 86.8 | 92.4 |
> | Qwen3-8B| 95.3 | 96.7|
> | PFT Unit Test Scripts| 4.53 | 4.27 |
> | VQT Unit Test Scripts| 4.57 | 4.23 |
> | Checklists | 4.87| 4.63 |
>
> The correctness rates consistently exceed 85%. VQT scripts show higher correctness as they involve fewer logic assertions. Interestingly, for lower-capability models (like Qwen3-8B), script correctness is higher (>95%) because the generated code is simpler, making the tests behave more predictably (often failing correctly). **These results indicate that the unit test executions are generally reliable, giving confidence that our automated evaluation accurately reflects model behavior.**
>
>
> #### 2. On Manual Inspection (Weakness 2)
> > **Weakness 2:** In line 259, the author mentioned "manual inspection and rule-based validation," but did not provide sufficient details on how the inspection was conducted or what specific rules were used for validation. In Appendix B, they stated that only 10 synthesized instances were randomly sampled for manual inspection, which makes this evaluation rather limited and less convincing.
>
> We appreciate the opportunity to clarify the procedures for manual inspection and checklist design.
>
> The "manual inspection" mentioned in Line 259 was conducted **during the benchmark construction process** and was not the same as the sampling described in Appendix B. This manual inspection referred to a comprehensive executable validity check for **all 150 problems'** PFT and VQT test scripts. Specifically, we performed the following two-stage process:
>
>   1. **Rule-Based Checks:** Automated scripts filtered out syntax errors, invalid API calls, and other basic execution issues.
>   2. **Human Review:** Two graduate-level annotators reviewed each script to fix execution errors (e.g., ensuring selectors matched the DOM). If the two annotators disagreed on a fix, a third expert arbitrator resolved the conflict. This ensured that the final benchmark suite only contained runnable, valid test scripts.
>
> In contrast, Appendix B described a separate, post-construction **random sampling of 10 (later expanded to 30) instances** for **manual verification of correctness**, assessing whether the scripts produced expected test behavior. The details of this sampling and correctness validation are described in the previous response.
>
> This distinction clarifies that manual inspection covered **all scripts for executability during benchmark construction**, while Appendix B was a **random correctness check** on a small subset for validation purposes. **We will explicitly note this distinction in the revised version to remove any ambiguity.**

---

> ### Author Response · Authors · 2025-11-25
>
> #### 3. On Checklist Design (Question 2)
>
> > **Question 2:** For VLM-as-Judge, how were the checklists designed? How was the manual inspection conducted, and how many evaluators were involved? Additionally, how were conflicts between human evaluators resolved?
>
> The VLM-as-Judge checklists were not arbitrary. As detailed in Appendix E (Figures 14 & 15), they were synthesized using strict prompts that enforced five key criteria:
>   1. **Output-Oriented:** Focus on the rendered visualization, not the UI styling.
>   2. **Input-Output Connection:** Explicitly link the control state to the visual result (e.g., "If slider X is 50%, graph Y must show...").
>   3. **Scientific Focus:** Prioritize scientific accuracy (curves, physics, equations).
>   4. **Visual Verification:** Ensure items are verifiable via a screenshot.
>   5. **Strict Adherence:** Do not hallucinate requirements outside the Implementation Plan.
>
>
> #### 4. On Counter-Intuitive Results for "Hard" Problems (Weakness 3)
> > **Weakness 3:**  In figure 3, the results show better Overall Pass Rate (OPR) on hard question, which is a little bit counter-intuitive. From the analysis, the author did not provide a good explanation of this.
>
> You correctly noted that the Overall Pass Rate (OPR) for PFT is higher for "Hard" problems (Figure 3), which seems counter-intuitive.
>
> **Explanation:**
>
> This phenomenon stems from how "Difficulty" and "OPR" are defined in our benchmark:
> - **Definition of Difficulty:** We categorize difficulty based on the **number of interactive components** (Easy: 1-3, Medium: 4-6, Hard: 7-10).
> - **Nature of PFT:** PFT measures **functional implementation compliance** (e.g., "Does the slider exist?", "Can the button be clicked?", "Is the slider step size correct?"). The OPR is the direct pass-rate indicator for PFT, reflecting how many of these functional checks the generated code satisfies.
>
> We think the cause of this phenomenon is the types of components that appear in problems labeled as "Hard" under our difficulty definition. When we examined the questions in physics domain manually, we found that "Hard" questions include a larger share of checkbox components (``<input type="checkbox">``), which have very simple binary logic. For example, the ten "Easy" questions contain 25 components in total, with only 2 checkboxes (8%). In contrast, the ten "Hard" questions contain 86 components, with 31 checkboxes (36%).
>
> In "Easy" questions, the most common component type is the slider (``<input type="range">``) (17/25). A slider requires parameters such as minimum and maximum values, step size, and default value, which makes its implementation more complex. "Medium" questions often contain even more complex controls, such as clickable elements inside a canvas.
>
> Because PFT measures functional implementation compliance, the presence of many simple components raises the OPR for Hard questions. However, simple code logic does not imply simple scientific interaction logic. A binary checkbox may still correspond to a complex scientific effect in the visualization canvas, such as whether to show reflected light based on underlying physical rules. This supports our decision to define difficulty by the number of components, since more components usually imply more scientific interactions. This also matches the pattern in Figure 3 where the difficulty levels defined by component count align with the distribution of VQT scores, where Hard tasks clearly show lower visual correctness.
>
> We will integrate this clarification into Section 5.2 so that readers understand why PFT increases with component count while scientific correctness becomes harder to achieve.

---

### Official Review · Reviewer_Y5KH · 2025-10-29

**Soundness:** 3
**Presentation:** 3
**Contribution:** 3
**Rating:** 6
**Confidence:** 4

**Summary:**

The authors present a new benchmark for language models that evaluates their ability to create interactive simulations of various mathematical, statistical, or scientific concepts by following the instructions laid out in a plan. The artifacts produced by the models are evaluated against reference demonstrations from the Wolfram Demonstrations Project, measuring fidelity in terms of both action-assertion tuples and the visual similarity of the outputs. The authors evaluate a large number of open- and closed-source LLMs on their benchmarks and provide a descriptive analysis of their results, arguing that the benchmark serves as an unsolved challenge which could be useful for developing more reliable AI assistants.

**Strengths:**

The primary strength of this paper is its clarity: the motivation is well-supported, the technical aspects are well-explained (for the most part -- see below), and the conclusions generally are well-supported by the data. I’m also pleased with the wide range of models that were examined and the additional sub-experiments performed to validate the various metrics used for evaluation (though some details are missing -- again see below). I think that the topic of the paper is both timely and relevant.

**Weaknesses:**

While the overall flow of the paper is clear, there are a few points in which important details are omitted or left vague which make the finer points more difficult to follow. For instance, the “Action Success Rate” is described as “the percentage of cases where the expected visual state appears after the specified action sequence” but “expected” is not clearly defined in either the main text or appendix. A reader might assume this refers to a scientifically correct output, but a later reference to ASR makes it seem as though it accounts for only the ability to, for instance, click a button or move a slider. In addition, unlike the other VQT measure, ASR is defined as an accuracy rate. If it’s based on the output of a VLM, how is the judge’s confidence incorporated into the measure?

In addition, I was not able to find any details about the human study used to validate the VLM-as-judge. How many participants were there, and how were they recruited / compensated? Again, I appreciate the addition of this experiment but it is difficult to assess its implications without more details.

Finally, I think the paper would benefit from an explicit example of the entire evaluation process (i.e. a sequence of screenshots showing the output after each action along with the corresponding image from the reference demonstration), which would help clarify some of the points raised above.

**Questions:**

- How is the Action Success Rate defined and computed?
- What are the full details for the human study used to validate the VLM-as-judge?
- Are there are common threads between the scientific concepts that models appear to struggle with (i.e. across disciplines)?

**Details Of Ethics Concerns:**

The authors include a human user study, but I wasn't able to find any details about the participants (i.e. number, recruitment process, compensation, or IRB approval).

---

> ### Author Response · Authors · 2025-11-25
>
> Dear Reviewer Y5KH,
>
> We sincerely thank you for your positive evaluation and for recognizing the INTERACTSCIENCE benchmark as timely and relevant. We appreciate your constructive feedback regarding the definitions of metrics and the details of our validation study. Below, we address your specific questions and weaknesses.
>
> #### 1. On Action Success Rate (ASR) Definition (Weakness 1 & Question 1)
>
> > **Weakness 1:** While the overall flow of the paper is clear, there are a few points in which important details are omitted or left vague which make the finer points more difficult to follow. For instance, the "Action Success Rate" is described as "the percentage of cases where the expected visual state appears after the specified action sequence" but "expected" is not clearly defined in either the main text or appendix. A reader might assume this refers to a scientifically correct output, but a later reference to ASR makes it seem as though it accounts for only the ability to, for instance, click a button or move a slider. In addition, unlike the other VQT measure, ASR is defined as an accuracy rate. If it's based on the output of a VLM, how is the judge's confidence incorporated into the measure?
> **Question 1:** How is the Action Success Rate defined and computed?
>
> You rightly pointed out the ambiguity in our definition of "Action Success Rate." We apologize for the confusion and clarify its exact definition and computation here.
>
> - **Definition:** ASR is a test pass rate, not a scientific correctness metric. It measures whether the model-generated code allows the automated test script (Playwright) to **execute the full sequence of user actions** (e.g., "click button," "drag slider") without crashing or timing out, effectively producing a final snapshot file.
> - **Computation:** ASR is computed in the same way as the rate metrics used in our programmatic functional tests. It is the proportion of cases in which the unit test script completes the full action sequence without errors. **It does not depend on any VLM output**, so the judge's confidence does not enter into its calculation. Among all our metrics, only the VLM score used in VQT depends on a vision-language model. For that part of the evaluation, we fix the judge model (Gemini-2.5-Pro) and set its temperature to 1.0 to keep the output stable across runs.
> - **Implication:** This metric explains the "Superficial Competence" gap we discuss in Section 5.2. Models have high ASR (>85%) because they can easily generate working UI widgets (buttons that click, sliders that slide), but the resulting visual state—captured in the snapshot—is often scientifically incorrect (low VLM score).
>
> We will revise Section 5.1 to explicitly define ASR as "Execution Success Rate" to prevent readers from conflating it with scientific accuracy.
>
>
> #### 2. On Human Study Details (Weakness 2 & Question 2)
>
> >**Weakness 2:** In addition, I was not able to find any details about the human study used to validate the VLM-as-judge. How many participants were there, and how were they recruited / compensated? Again, I appreciate the addition of this experiment but it is difficult to assess its implications without more details.
> **Question 2:** What are the full details for the human study used to validate the VLM-as-judge?
>
> Thank you for pointing out the missing details of our human study. We clarify the full experimental setup below, and will add these to the revised paper.
> - **Sampling Strategy:**
> We sampled **30 problems** stratified across 5 disciplines and 3 difficulty levels (one per discipline–difficulty pair). For each problem, outputs were collected from three models of different capability levels: Gemini-2.5-Pro, DeepSeek-V3, and Qwen3-8B, producing **90 samples** in total. Each model output produced four screenshots, and each screenshot was paired with one checklist. On average, each checklist contained 5.75 scoring items.
> - **Annotator Recruitment and Protocol:**
> We recruited **5 graduate students** as annotators. They were compensated based on the number of problems annotated, receiving one US dollar for every five problems. Each  was independently scored by **3 annotators**, the exact same checklist-based scoring rubric (1-5 scale) provided to the VLM-Judge. Annotators were compensated at a standard hourly rate consistent with our institution's student research assistant guidelines. The final human score for each sample was the mean of the three annotations.
> - **Inter-Annotator Agreement:**
> We computed Krippendorff’s alpha across annotators and obtained **0.7326**, showing stable agreement for our human ground truth.
> - **Validation Result:**
> The VLM-Judge (Generated Snapshot + Reference + Checklist) achieved a Spearman correlation of **0.8827** with the human ground truth. We will include the associated p-values and MAE in the final revision for completeness.

---

> ### Author Response · Authors · 2025-11-25
>
> #### 3. On Common Failure Threads (Question 3)
>
> > **Question 3:** Are there common threads between the scientific concepts that models appear to struggle with (i.e. across disciplines)?
>
> You asked if there are common failure threads across disciplines.
> **UI-Logic Disconnect**:
> The most persistent failure across all disciplines (Physics, Math, Earth Science) is the inability to bind interaction events to scientific logic.
>   - **Observation:** Models treat the Implementation Plan as two separate tasks: "Build a UI" and "Draw a Picture." They rarely succeed in connecting them.
>   - **Example:** In Physics (Figure 6b), the model draws a spring and sliders perfectly (High ASR/CLIP), but when the slider moves, the spring distorts incorrectly because the underlying physics formula wasn't updated in the JavaScript event listener.
>   - **Cross-Discipline Pattern:** Whether it is updating a Huffman tree in Computer Science or a planetary orbit in Earth Science, models consistently struggle to maintain the **state dependency** between the HTML control and the Canvas rendering. They prioritize visual style over logical function.
>
> #### 4. On Visual Examples (Weakness 3)
>
> > **Weakness 3:** Finally, I think the paper would benefit from an explicit example of the entire evaluation process (i.e. a sequence of screenshots showing the output after each action along with the corresponding image from the reference demonstration), which would help clarify some of the points raised above.
>
> We fully agree that an explicit, step-by-step visual walkthrough of a single test case is essential for clarifying the evaluation process and metrics like ASR.
>
> We note that Appendix Figures 7 and 8 currently illustrate the visual results of the VQT process by displaying the different screenshot states that are captured for evaluation.
>
> To provide the requested clarity, we will add a new, dedicated figure that visualizes the complete lifecycle of an evaluation test case, integrating all components of our hybrid framework:
>   1. **Plan Segment:** An excerpt from the Implementation Plan defining the expected interaction (e.g., the plan of a demo on "Wave function and simple harmonic motion in transverse waves").
>   2. **Action Sequence:** A sequence of screenshots demonstrating the automated Playwright script moving the control element (e.g., dragg the slider for wave with increased amplitude and frequency, and phase shifted to π/3).
>   3. **Generated Output vs. Reference:** The final generated visualization compared directly against the reference demonstration's screenshot for that state.
>   4. **Evaluation Snapshot:** The resulting failure (e.g., The curve shape is correct, but the amplitude did not change), coupled with the failing Checklist item and the final VLM score.
>
> In addition, we have added an **``examples``** folder in the **Supplementary Material**, which contains all the components and results for the case illustrated in Appendix Figure 8. This includes the **problem description, checklist, PFT and VQT test scripts, and the resulting HTML files**. These additions aim to provide full transparency and allow readers to inspect the complete evaluation workflow.
>
> We believe these additions will significantly improve the presentation and clarity of our evaluation methodology.

---

### Official Review · Reviewer_uoa9 · 2025-10-31

**Soundness:** 3
**Presentation:** 2
**Contribution:** 2
**Rating:** 2
**Confidence:** 5

**Summary:**

This paper presents scientific demonstration code generation by large language models. Towards that end, the authors adapted VLM-as-judge to evaluate the tasks.

**Strengths:**

- Timely topic focusing on the LLM and code generation
- Broader impact towards the research community

**Weaknesses:**

1. The human evaluation of VLM-as-Judge is absent. I recommend the authors to consult the following paper [1], specifically Table G1 and G2 of Appendix with human judgment and LLM-as-Judge correlation and Cohen’s Kappa between multiple raters on a small subset. This step is crucial, as LLMs demonstrate hallucinations and prompting language also determines LLM evaluation quality.

2. The recent relevant literature is missing. In fact, SWE-bench Multimodal already have multimodal programming questions answered [2].



Reference

1. Zhou, Xuhui, Hyunwoo Kim, Faeze Brahman, Liwei Jiang, Hao Zhu, Ximing Lu, Frank Xu et al. "Haicosystem: An ecosystem for sandboxing safety risks in human-ai interactions." COLM 2025
https://arxiv.org/pdf/2409.16427
2. Yang, J., Jimenez, C.E., Zhang, A.L., Lieret, K., Yang, J., Wu, X., Press, O., Muennighoff, N., Synnaeve, G., Narasimhan, K.R. and Yang, D., SWE-bench Multimodal: Do AI Systems Generalize to Visual Software Domains?. In The Thirteenth International Conference on Learning Representations. https://www.swebench.com/multimodal.html

**Questions:**

How does the proposed benchmark perform compared to the SWE-bench Multimodal?
Link:  https://www.swebench.com/multimodal.html

---

> ### Author Response · Authors · 2025-11-25
>
> Dear Reviewer uoa9,
>
> We sincerely thank you for your review and for recognizing the "timely" nature of this topic and its broader impact on the research community.
> #### 1. On Human Evaluation of VLM-as-Judge (Weakness 1)
> > **Weakness 1:** The human evaluation of VLM-as-Judge is absent. I recommend the authors to consult the following paper [1], specifically Table G1 and G2 of Appendix with human judgment and LLM-as-Judge correlation and Cohen's Kappa between multiple raters on a small subset. This step is crucial, as LLMs demonstrate hallucinations and prompting language also determines LLM evaluation quality.
>
> We fully agree that validating the VLM-as-Judge against human judgment is important to avoid hallucinations and ensure reliability. We appreciate your suggestion to include inter-rater reliability metrics like Cohen's Kappa.
>
> **Clarification of Human Evaluation Methodology:**
> In the submitted version, we conducted a human validation study to derive the correlations reported in Table 3. To address your concern about the robustness of this study, we will expand the paper to include the following detailed experimental setup:
> - **Sampling Strategy:** We randomly sampled **30 problems** stratified across all 5 disciplines and 3 difficulty levels (selecting one problem per discipline-difficulty pair). For each problem, we collected outputs from three distinct models representing different capability tiers: Gemini-2.5-Pro, DeepSeek-V3, and Qwen3-8B. This resulted in a total of **90 samples** for human evaluation.
> - **Annotation Protocol:** We recruited **5 graduate students** as expert annotators. Each sample was independently evaluated by **3 annotators** using the exact same checklist-based scoring rubric (1-5 scale) provided to the VLM-Judge. The final human ground truth was derived from the mean score of the three annotators.
> - **Inter-Annotator Agreement:** To ensure the quality of our human ground truth, we calculated **Krippendorff's alpha** for the annotators, achieving a score of **0.7326**. This indicates a acceptable high level of agreement among human experts, validating the quality of the ground truth used for validating the VLM judge.
>
> **Our proposed VLM-Judge configuration (Generated Snapshot + Reference + Checklist) achieved a Spearman correlation of 0.8827 with this high-quality human ground truth, demonstrating strong alignment.**
>
> In the final version, we will:
>   1. Add the **Krippendorff's alpha** score to the main text to demonstrate human reliability.
>   2. Update **Table 3** to include the p-values for Spearman correlations to demonstrate statistical significance.
>
> | Judge Input | Corr. | Corr. p-value |
> |-|-|-|
> | $I _{\mathrm{gen}}$ $+$ $I _{\mathrm{ref}}$ $+$ $L$ | 0.8827 | $1.23 \times 10^{-26}$ |
> | $I _{\mathrm{gen}}$ $+$ $L$ | 0.8224 | $5.67 \times 10^{-23}$ |
> | $I _{\mathrm{gen}}$ $+$ $I _{\mathrm{ref}}$ | 0.3837 | $1.01 \times 10^{-19}$ |
> | $I _{\mathrm{gen}}$ | 0.7360  | $9.87 \times 10^{-27}$ |
> | $C$ | 0.1408 | $3.21 \times 10^{-6}$ |
>
> We believe these additional details and metrics will fully address your concern regarding the rigorous validation of our VLM-as-Judge framework.

---

> ### Author Response · Authors · 2025-11-25
>
> #### 2. On Comparison with SWE-bench Multimodal (Weakness 2 & Question)
> >**Weakness 2**: The recent relevant literature is missing. In fact, SWE-bench Multimodal already have multimodal programming questions answered [2].
> **Question:** How does the proposed benchmark perform compared to the SWE-bench Multimodal?
>
> We thank you for pointing out SWE-bench Multimodal. It is indeed a relevant benchmark on multimodal programming. However, our INTERACTSCIENCE occupies a distinct niche focusing on scientific demonstration code generation rather than software maintenance.
>
> **Key Differences:**
> 1. **Different Tasks with Clear Functional Focus:** SWE-bench Multimodal focuses on **repository-level bug fixing** (resolving GitHub issues) where visual context helps identify the bug. INTERACTSCIENCE focuses on **scientific demonstration code generation** (converting a theorem/concept into an interactive webpage in HTML+CSS+JS).
> 2. **Stronger Scientific Reasoning Requirements:** While both our benchmark and SWE-bench Multimodal examine the model's capacity to align code with visual elements (e.g., verifying interactive mapping and visual consistency between code and questions/images), our evaluation goes a step further. We place additional emphasis on **ensuring the scientific correctness of generated outputs**. For instance, verifying whether a physics simulation accurately adheres to scientific laws. As a result, our benchmark demands stronger **scientific reasoning** capabilities.
> 3. **Novel Evaluation Method Combining Code Tests and Visual Checks:** SWE-bench and SWE-bench Multimodal rely on existing test suites only. INTERACTSCIENCE introduces a hybrid generative evaluation. We synthesize not only programmatic test suites for functional interaction logic, but also visually-grounded qualitative testing with LLM-as-judge and reference snapshots for checking visually-grounded scientific correctness. **This hybrid design forms a new evaluation approach that supports both functional reliability and scientific validity in generated interactive scientific code.**
>
> To clearly position our contribution, we have prepared the following comparison table which we will include in the final version.
>
> | Benchmark | Primary Task | Scientific Reasoning | Interaction-based Testing | Visualization-based Testing | Evaluation |
> |---|---|---|---|---|---|
> | **Scientific Visualization** |  |  |  |  |  |
> | SridBench | Scientific Visualization Generation | ✓ | ✗ | ✓ | LLM-judge |
> | EduVisBench | Scientific Visualization Generation | ✓ | ✗ | ✓ | LLM-judge |
> | **Software Issues Resolving** |  |  |  |  |  |
> | SWE-bench | Repository-level Bug Fixing | ✗ | ✓ | ✗ | Automation |
> | SWE-bench Multimodal | Repository-level Bug Fixing with Visual Context | ✗ | ✓ | ✗ | Automation |
> | **Code Generation** |  |  |  |  |  |
> | LiveCodeBench | Competitive-Programming Code Generation | ✗ | ✗ | ✗ | Automation |
> | Interaction2Code | Interactive Web Page Generation | ✗ | ✗ | ✗ | Automation |
> | WebGen-Bench | Interactive Web Page Generation | ✗ | ✗ | ✗ | LLM-judge |
> | WebDev Arena | Web Design | ✗ | ✗ | ✓ | Human Voting |
> | ArtifactsBench | Interactive Visual Artifacts | ✗ | ✗ | ✓ | LLM-judge |
> | InteractScience | Interactive Scientific Demonstration Generation | ✓ | ✓ | ✓ | Automation + LLM-judge |
>
> As shown above, INTERACTSCIENCE is unique in requiring the intersection of **Scientific Reasoning**, **Interaction**, and **Visualization**, filled by our specific PFT+VQT framework.
>
> We hope this clarification and the comparative analysis address your concerns. We will add these references and the table into the final version.

---

### Meta-Review · Area_Chair_RcJW · 2026-01-06

**Summary:**

This paper proposes a benchmark dataset about code generation for implementing interactive scientific demonstrations. Most concerns are about the evaluation metrics accompanying the dataset:
1. Validity/reliability of the evaluation method: Reviewers are concerned about the use of VLM-as-judge and the insufficient human evaluation to validate it. Similarly, there's also a concern (from Reviewer df5Z) about the minimal human involvement in the test case validation process.
2. Lack of clarity or justification: Reviewers have questions and concerns about the specific evaluation metrics (Action Success Rate, Overall Pass Rate) and the complete evaluation process.
3. Missing failure analysis or insights about existing models.
4. Finally, reviewers (uoa9, ei4T) questioned the novelty and broad impact of the benchmark.

**Reviewer Concerns:**

To these concerns,
1. The authors emphasized Table 3 (showing the correlation between the VLM metric and the human expert score) as key evidence of the validity of the VLM judge. They further added a human evaluation based on 30 sampled problems to validate the quality of the ground truth used to validate the VLM judge.
Similarly, to address the concern about limited human involvement in test case construction, the authors added a human evaluation on 30 sampled questions and reported the faithfulness and correctness of the test cases. They also clarified the details of manual inspection and rule-based validation.
2.  The authors clarified the definitions and intents of the two metrics.
3. Analysis is added in response.
4. A table comparing the proposed benchmark with prior ones is added.

**Reviewer Scores:**

While other concerns were addressed well, the concern about evaluation metrics and the dataset quality may still exist and the reviewers may not increase their scores.

Specifically, about the dataset quality, the authors did not provide the annotation guidelines for evaluating the faithfulness and correctness of the test cases. It is also notable that, under a scale of 1-5, the correctness of unit test scripts is merely 4.1-4.2, and the faithfulness of the test cases is 4.4-4.5. The authors did not interpret and justify these ratings, which renders the dataset quality questionable.

About the evaluation metrics, I'm a little concerned that the metrics do not align with the critical capability of models in performing the proposed code generation task. For example, as raised by Reviewer df5Z, models are able to achieve higher Overall Pass Rates on harder questions, which is counterintuitive. Based on the authors' response, this is caused by the fact that this metric can be easily confused by trivial functional implementations (e.g., checkbox implementations). An improved metric design may be expected by the reviewers.

---

### Decision · Program_Chairs · 2026-01-26

Reject